# Understanding Curriculum Learning in Policy Optimization for Online Combinatorial Optimization

**Runlong Zhou**                                            *vectorzh@cs.washington.edu*
*University of Washington*

**Zelin He**                                                       *zelinh2@uw.edu*
*University of Washington*

**Yuandong Tian**                                            *yuandong@fb.com*
*Facebook AI Research*

**Yi Wu**                                                          *jxwuyi@gmail.com*
*Tsinghua University*

**Simon S. Du**                                              *ssdu@cs.washington.edu*
*University of Washington*

**Reviewed on OpenReview:** *https: // openreview. net/ forum? id= gKEbBKRUjA*

## Abstract

Over the recent years, reinforcement learning (RL) starts to show promising results in tackling combinatorial optimization (CO) problems, in particular when coupled with curriculum learning to facilitate training. Despite emerging empirical evidence, theoretical study on why RL helps is still at its early stage. This paper presents the first systematic study on policy optimization methods for online CO problems. We show that online CO problems can be naturally formulated as latent Markov Decision Processes (LMDPs), and prove convergence bounds on natural policy gradient (NPG) for solving LMDPs. Furthermore, our theory explains the benefit of curriculum learning: it can find a strong sampling policy and reduce the distribution shift, a critical quantity that governs the convergence rate in our theorem. For a canonical online CO problem, the Best Choice Problem (BCP), we formally prove that distribution shift is reduced *exponentially* with curriculum learning *even if the curriculum is a randomly generated BCP on a smaller scale*. Our theory also shows we can simplify the curriculum learning scheme used in prior work from multi-step to single-step. Lastly, we provide extensive experiments on the Best Choice Problem, Online Knapsack, and AdWords to verify our findings.

## 1 Introduction

In recent years, machine learning techniques have shown promising results in solving combinatorial optimization (CO) problems, including traveling salesman problem (TSP, Kool et al. (2019)), maximum cut (Khalil et al., 2017) and satisfiability problem (Selsam et al., 2019). While in the worst case some CO problems are NP-hard, in practice, the probability that we need to solve the worst-case problem instance is low (Cappart et al., 2021). Machine learning techniques are able to find generic models which have exceptional performance on the majority of a class of CO problems.

A significant subclass of CO problems is called online CO problems, which has gained much attention (Grötschel et al., 2001; Huang, 2019; Garg et al., 2008). Online CO problems entail a sequential decision-making process, which perfectly matches the nature of reinforcement learning (RL).

This paper concerns using RL to tackle online CO problems. RL is often coupled with specialized techniques including (a particular type of) Curriculum Learning (Kong et al., 2019), human feedback and correction (Pérez-Dattari et al. (2018), Scholten et al. (2019)), and policy aggregation (boosting, Brukhim et al. (2021)). Practitioners use these techniques to accelerate the training speed.

While these hybrid techniques enjoy empirical success, the theoretical understanding is still limited: it is unclear when and why they improve the performance. In this paper, we particularly focus on *RL with Curriculum Learning* (Bengio et al. (2009), also named "bootstrapping" in Kong et al. (2019)): train the agent from an easy task and *gradually* increase the difficulty until the target task. Interestingly, these techniques exploit the special structures of online CO problems.

**Main contributions.** In this paper, we initiate the formal study on using RL to tackle online CO problems, with a particular emphasis on understanding the specialized techniques developed in this emerging subarea. Our contributions are summarized below.

• **Formalization.** For online CO problems, we want to learn a single policy that enjoys good performance over *a distribution* of problem instances. This motivates us to use Latent Markov Decision Process (LMDP) (Kwon et al., 2021a) instead of standard MDP formulation. We give concrete examples, *the Best Choice Problem* (BCP, also known as *the Secretary Problem*), *Online Knapsack*, and *AdWords* (*Online Matching and Ad Allocation*, ADW), to show how LMDP models online CO problems. With this formulation, we can systematically analyze RL algorithms.

• **Provable efficiency of policy optimization.** By leveraging recent theory on Natural Policy Gradient for standard MDP Agarwal et al. (2021), we analyze the performance of NPG for LMDP. The performance bound is characterized by the number of iterations, the excess risk of policy evaluation, the transfer error, and the relative condition number $\kappa$ that characterizes the distribution shift between the sampling policy and the optimal policy. We also take into account the effect of entropy regularization. To our knowledge, this is the first performance bound of policy optimization methods on LMDP.

• **Understanding and simplifying Curriculum Learning.** Using our performance guarantee on NPG for LMDP, we study when and why Curriculum Learning is beneficial to RL for online CO problems. Our main finding is that the main effect of Curriculum Learning is to give *a stronger sampling policy*. Under certain circumstances, Curriculum Learning reduces the relative condition number $\kappa$, improving the convergence rate. For BCP, we provably show that Curriculum Learning can *exponentially reduce $\kappa$* compared with using the naïve sampling policy. Surprisingly, this means *even a randomly constructed curriculum of BCP accelerates the training exponentially*. As a direct implication, we show that the multi-step Curriculum Learning proposed in Kong et al. (2019) can be significantly *simplified into a single-step scheme*. Lastly, to obtain a complete understanding, we study the failure mode of Curriculum Learning, in a way to help practitioners to decide whether to use Curriculum Learning based on their prior knowledge. To verify our theories, we conduct extensive experiments on three classical online CO problems [BCP, Online Knapsack (decision version, OKD), and ADW (decision version)] and carefully track the dependency between the performance of the policy and $\kappa$.

## 2    Related Works

**Combinatorial Optimization problems.** CO has been a long lasting field of people's interest. There are a rich literature regarding CO problems such as traveling salesman problem (Flood (1956); Bellmore & Nemhauser (1968)), maximum cut (Karp (1972); Goemans & Williamson (1995)), and satisfiability problem (Cook (1971); Trakhtenbrot (1984)).

**RL for CO.** There have been rich literature studying RL for CO problems, e.g., using Pointer Network in REINFORCE and Actor-Critic for routing problems (Nazari et al., 2018), combining Graph Attention Network with Monte Carlo Tree Search for TSP (Drori et al., 2020) and incorporating Structure-to-Vector Network in Deep Q-networks for maximum independent set problems (Cappart et al., 2019). Bello et al. (2017) proposed a framework to tackle CO problems using RL and neural networks. Kool et al. (2019) combined REINFORCE and attention technique to learn routing problems. Vesselinova et al. (2020) and Mazyavkina et al. (2021) are taxonomic surveys of RL approaches for graph problems. Bengio et al. (2020)

summarized learning methods, algorithmic structures, objective design and discussed generalization. In particular scaling to larger problems was mentioned as a major challenge. Compared to supervised learning, RL not only mimics existing heuristics, but also discover novel ones that humans have not thought of, for example chip design (Mirhoseini et al., 2021) and compiler optimization (Zhou et al., 2020b). Theoretically, there is a line of work on analyzing data-driven approach to combinatorial problems (Balcan, 2020). However, to our knowledge, the theoretical analysis for RL-based method is still missing.

Kong et al. (2019) focused on using RL to tackle *online* CO problems, which means that the agent must make sequential and irrevocable decisions. They encoded the input in a length-independent manner. For example, the $i$-th element of a $n$-length sequence is encoded by the fraction $i/n$ and other features, so that the agent can generalize to unseen $n$, paving the way for Curriculum Learning. Three online CO problems were mentioned in their paper: ADW, Online Knapsack, and BCP. Currently, Online Matching (ADW) and Online Knapsack have only approximation algorithms (Huang et al., 2019; Albers et al., 2021). There are also other works about RL for online CO problems. Alomrani et al. (2021) uses deep-RL for Online Matching. Oren et al. (2021) studies Parallel Machine Job Scheduling problem (PMSP) and Capacitated Vehicle Routing problem (CVRP), which are both online CO problems, using offline-learning and Monte Carlo Tree Search.

**LMDP.** We provide the exact definition of LMDP in Section 4.1. As studied by Steimle et al. (2021), in the general cases, optimal policies for LMDPs are *history-dependent*. This is different from standard MDP cases where there always exists an optimal *history-independent* policy. They showed that even finding the optimal history-independent policy is *NP-hard*. Kwon et al. (2021a) investigated the sample complexity and regret bounds of LMDP in the history-independent policy class. They presented an exponential lower-bound for a general LMDP and derived algorithms with polynomial sample complexities for cases with special assumptions. Kwon et al. (2021b) showed that in reward-mixing MDPs, where MDPs share the same transition model, a polynomial sample complexity is achievable without any assumption to find an optimal history-independent policy.

**Convergence rate for policy gradient methods.** There is line of work on the convergence rates of policy gradient methods for standard MDPs (Bhandari & Russo (2021), Wang et al. (2020), Liu et al. (2020), Ding et al. (2021), Zhang et al. (2021)). For *softmax tabular parameterization*, NPG can obtain an $O(1/T)$ rate (Agarwal et al., 2021) where $T$ is the number of iterations; with entropy regularization, both PG and NPG achieves linear convergence (Mei et al., 2020; Cen et al., 2021). For *log-linear policies*, sample-based NPG makes an $O(1/\sqrt{T})$ convergence rate, with assumptions on $\epsilon_{\text{stat}}, \epsilon_{\text{bias}}$ and $\kappa$ (Agarwal et al., 2021) (see our Definition 4); exact NPG with entropy regularization enjoys a linear convergence rate up to $\epsilon_{\text{bias}}$ (Cayci et al., 2021). We extend the analysis to LMDP.

**Curriculum Learning.** There are a rich body of literature on Curriculum Learning (Zhou et al., 2021b;a; 2020a; Ao et al., 2021; Willems et al., 2020; Graves et al., 2017). As surveyed in Bengio et al. (2009), Curriculum Learning has been applied to training deep neural networks and non-convex optimizations and improves the convergence in several cases. Narvekar et al. (2020) rigorously modeled curriculum as a directed acyclic graph and surveyed work on curriculum design. Kong et al. (2019) proposed a bootstrapping (Curriculum Learning) approach: gradually increase the problem size after the model works sufficiently well on the current problem size.

## 3 Motivating Online CO Problems

Online CO problems are a natural class of problems that admit constructions of small-scale instances, because the hardness of them can be characterized by the input length, and instances of different scales are similar. This property simplifies the construction of curricula and underscores curriculum learning. We also believe online CO problems make the use of LMDP suitable, because under a proper distribution $\{w_m\}$, *instances in a large portion of the probability space have similar near optimal solutions.*

In this section we introduce three motivating online CO problems. We are interested in these problems because they have all been extensively studied. Furthermore, they were studied in Kong et al. (2019), the

paper that motivates our work. They also have real-world applications, e.g., auction design (Babaioff et al., 2007) and advertisement targeting (Mehta et al., 2007).

### 3.1 The Best Choice Problem (BCP)[1]

In BCP, the goal is to maximize the *probability* of choosing the maximum among $n$ *different* numbers, where $n$ is known. They arrive sequentially and when the $i$-th number shows up, the decision-maker observes the relative ranking $X_i$ among the first $i$ numbers, which means being the $X_i$th-best so far. A decision that whether to accept or reject the $i$-th number must be made *immediately* when it comes, and such decisions *cannot be revoked*. Once one number is accepted, the game ends immediately.

The ordering of the numbers is unknown. There are in total $n!$ permutations, and an instance of BCP is drawn from an unknown distribution over these permutations. In the classical BCP, each permutation is sampled with equal probability. The optimal solution for the classical BCP is the well-known $1/e$-*threshold strategy*: reject all the first $\lfloor n/e \rfloor$ numbers, then accept the first one which is the best so-far. In this paper, we also study some different distributions.

### 3.2 Online Knapsack (decision version, OKD)

In Online Knapsack problems the decision-maker observes $n$ (which is known) items arriving sequentially, each with value $v_i$ and size $s_i$ revealed upon arrival. A decision to either accept or reject the $i$-th item must be made *immediately* when it arrives, and such decisions *cannot be revoked*. At any time the accepted items should have their total size no larger than a known budget $B$.

The goal of standard Online Knapsack is to maximize the total value of accepted items. In this paper, we study its decision version, whose goal is to maximize the *probability* of total value reaching *a known target* $V$.

We assume that all values and sizes are sampled independently from two fixed distributions, namely $v_1, v_2, \ldots, v_n \overset{\text{i.i.d.}}{\sim} F_v$ and $s_1, s_2, \ldots, s_n \overset{\text{i.i.d.}}{\sim} F_s$. In Kong et al. (2019) the experiments were carried out with $F_v = F_s = \text{Unif}_{[0,1]}$, and we also study other distributions.

**Remark 1.** A challenge in OKD is the sparse reward: the only signal is reward 1 when the total value of accepted items first exceeds $V$ (see the detailed formulation in Appendix C.2), unlike in Online Knapsack the reward of $v_i$ is given instantly after the $i$-th item is successfully accepted. This makes random exploration hardly get reward signals, necessitating Curriculum Learning.

### 3.3 AdWords (decision version, ADW)

In ADW, there are $n$ advertisers each with budget 1 and $m$ ad slots. Each ad slot $j$ arrives sequentially along with a vector $(v_{1,j}, v_{2,j}, \ldots, v_{n,j})$ where $v_{i,j}$ is the value that advertiser $i$ wants to pay for ad slot $j$. Once an ad slot arrives, it must be *irrevocably* allocated to an advertiser or not allocated at all. If ad slot $j$ is allocated to advertiser $i$ and the remaining budget of advertiser $i$ is not less than $v_{i,j}$, the total revenue increases by $v_{i,j}$ while advertiser $i$'s budget decreases by $v_{i,j}$.

We assume that for any advertiser $i$, $v_{i,1}, v_{i,2}, \ldots, v_{i,m} \overset{\text{i.i.d.}}{\sim} F_i$. Kong et al. (2019) studied a very special case called online $b$-matching where $F_i$ is a Bernoulli distribution. We study different distributions.

The objective of the standard ADW is to maximize the total revenue. For a similar reason as in OKD, we set *a known target* $V$ for the decision version. The goal of ADW is to maximize the *probability* of total revenue reaching $V$.

---

[1]We follow the statement in Kong et al. (2019) that BCP (secretary problem) is a CO problem. It is categorized as an optimal stopping problem.

# 4 Problem Setup

In this section, we first introduce LMDP and why it naturally formulates online CO problems. Then we list necessary components required by Natural Policy Gradient for LMDP (Algorithm 1).

**Notations.** For any positive integer $n$, we denote $[n] := \{1, 2, \ldots, n\}$. For any vector $x \in \mathbb{R}^n$, we denote $x^{\otimes} := x \otimes x = xx^\top$ as the self-outer-product of $x$. Further for any $y \in \mathbb{R}^m$, we denote $(x \circ y)(i, j) := x(i)y(j)$.

## 4.1 Latent Markov Decision Process

Tackling an online CO problem entails handling a family of problem instances, and each instance can be modeled as a Markov Decision Process. For online CO problems, we want to find one algorithm that works for a family of problem instances and performs well on average over an (unknown) distribution over this family. To this end, we adopt the concept of Latent MDP which naturally models online CO problems.

Latent MDP (Kwon et al., 2021a) is a collection of MDPs $\mathcal{M} = \{\mathcal{M}_1, \mathcal{M}_2, \ldots, \mathcal{M}_M\}$. All the MDPs share state set $\mathcal{S}$, action set $\mathcal{A}$ and horizon $H$. Each MDP $\mathcal{M}_m = (\mathcal{S}, \mathcal{A}, H, \nu_m, P_m, r_m)$ has its own initial state distribution $\nu_m \in \Delta(\mathcal{S})$, transition $P_m : \mathcal{S} \times \mathcal{A} \to \Delta(\mathcal{S})$ and reward $r_m : \mathcal{S} \times \mathcal{A} \to [0, 1]$, where $\Delta(\mathcal{S})$ is the probability simplex over $\mathcal{S}$. Let $w_1, w_2, \ldots, w_M$ be the mixing weights of MDPs such that $w_m > 0$ for any $m$ and $\sum_{m=1}^{M} w_m = 1$. At the start of every episode, one MDP $\mathcal{M}_m \in \mathcal{M}$ is randomly chosen with probability $w_m$.

Due to the time and space complexities of finding the optimal history-dependent policies, we stay in line with Kong et al. (2019) and care only about finding the optimal *history-independent* policy. Let $\Pi = \{\pi : \mathcal{S} \to \Delta(\mathcal{A})\}$ denote the class of all the history-independent policies.

**Log-linear policy.** Let $\phi : \mathcal{S} \times \mathcal{A} \to \mathbb{R}^d$ be a feature mapping function where $d$ denotes the dimension of feature space. Assume that $\|\phi(s, a)\|_2 \leqslant B$. A log-linear policy is of the form:

$$\pi_\theta(a|s) = \frac{\exp(\theta^\top \phi(s, a))}{\sum_{a' \in \mathcal{A}} \exp(\theta^\top \phi(s, a'))}, \text{ where } \theta \in \mathbb{R}^d.$$

**Remark 2.** Log-linear parameterization is a generalization of *softmax tabular parameterization* by setting $d = |\mathcal{S}||\mathcal{A}|$ and $\phi(s, a) = \text{One-hot}(s, a)$. They are "scalable": if $\phi$ extracts important features from different $\mathcal{S} \times \mathcal{A}$s with a fixed dimension $d \ll |\mathcal{S}||\mathcal{A}|$, then a single $\pi_\theta$ can generalize.

**Value function, Q-function and advantage function.** The expected reward of executing $\pi$ on $M_m$ is defined via value functions. We incorporate *entropy regularization* for completeness because prior works (especially empirical works) used it to facilitate training. *Due to space limit, we defer all the entropy regularized notations, algorithm and theorem to Appendix A.* We define the value function:

$$V_{m,h}^\pi(s) := \mathbb{E}_{\mathcal{M}_m, \pi}\left[\sum_{t=0}^{h-1} r_m(s_t, a_t) \,\middle|\, s_0 = s\right],$$

where the expectation is with respect to the randomness of trajectory induced by $\pi$ in $\mathcal{M}_m$. Denote $V^\pi := \sum_{m=1}^{M} w_m \sum_{s_0 \in \mathcal{S}} \nu_m(s_0) V_{m,H}^\pi(s_0)$, then we need to find $\pi^\star = \arg\max_{\pi \in \Pi} V^\pi$. Denote $V^\star := V^{\pi^\star}$.

The Q-function can be defined in a similar manner:

$$Q_{m,h}^\pi(s, a) := \mathbb{E}_{\mathcal{M}_m, \pi}\left[\sum_{t=0}^{h-1} r_m(s_t, a_t) \,\middle|\, (s_0, a_0) = (s, a)\right],$$

and the advantage function is defined as $A_{m,h}^\pi(s, a) := Q_{m,h}^\pi(s, a) - V_{m,h}^\pi(s)$.

**Modeling BCP.** For BCP, each instance is a permutation of length $n$, and in each round an instance is drawn from an unknown distribution over all permutations. In the $i$-th step for $i \in [n]$, the state encodes the $i$-th number and its relative ranking so far. The transition is deterministic according to the problem

definition. A reward of 1 is given *if and only if* the maximum is accepted. We model the distribution as follows: for the $i$-th number, it has probability $P_i$ to be the best so-far and is independent of other $i'$. Hence, the weight of each instance is simply the product of the probabilities on each position. The classical BCP satisfies $P_i = 1/i$.

**Modeling OKD.** For OKD, each instance is a sequence of items with values and sizes drawn from unknown distributions $F_v$ and $F_s$. In the $i$-th step for $i \in [n]$, the state encodes the information of $i$-th item's value and size, the remaining budget, and the remaining target value to fulfill. The transition is also deterministic according to the problem definition, and a reward of 1 is given *if and only if* the agent obtains the target value for the first time. $F_v = F_s = \text{Unif}_{[0,1]}$ in Kong et al. (2019).

**Modeling ADW.** For ADW, each instance is a $n \times m$ matrix $(v_{i,j})_{(i,j) \in [n] \times [m]}$, with each row $i$ subject to a distribution $F_i$. In the $j$-th step for $j \in [m]$, the state encodes the value vector $(v_{1,j}, v_{2,j}, \ldots, v_{n,j})$, the remaining budget vector $(B_1, B_2, \ldots, B_n)$, and the remaining target revenue to fulfill. The transition is also deterministic according to the problem definition, and a reward of 1 is given *if and only if* the agent obtains the target revenue for the first time.

### 4.2 Algorithm components

In this subsection we will introduce some necessary notations used by our main algorithm.

**Definition 1** (Visitation Distribution). *The state visitation distribution and state-action visitation distribution at step $h \geqslant 0$ with respect to $\pi$ in $\mathcal{M}_m$ are defined as*

$$d_{m,h}^{\pi}(s) := \mathbb{P}_{\mathcal{M}_m, \pi}(s_h = s),$$
$$d_{m,h}^{\pi}(s, a) := \mathbb{P}_{\mathcal{M}_m, \pi}(s_h = s, a_h = a).$$

We will encounter a grafted distribution $\widetilde{d}_{m,h}^{\pi}(s, a) = d_{m,h}^{\pi}(s) \circ \text{Unif}_{\mathcal{A}}(a)$ which in general is not the state-action visitation distribution with respect to any policy. However, it can be attained by first acting under $\pi$ for $h$ steps to get states then sample actions from the uniform distribution $\text{Unif}_{\mathcal{A}}$. This distribution will be useful when we apply a variant of NPG, where the sampling policy is fixed.

Denote $d_{m,h}^{\clubsuit} := d_{m,h}^{\pi^{\clubsuit}}$ and $d^{\clubsuit}$ as short for $\{d_{m,h}^{\clubsuit}\}_{1 \leqslant m \leqslant M, 0 \leqslant h \leqslant H-1}$, here $\clubsuit$ can be any symbol.

We also need the following definitions for NPG, which are different from the standard versions for discounted MDP because weights $\{w_m\}$ must be incorporated in the definitions to deal with LMDP. In the following definitions, let $v$ be the collection of any distribution, which will be instantiated by $d^{\star}$, $d^t$, etc. in the remaining sections.

**Definition 2** (Compatible Function Approximation Loss). *Let $g$ be the parameter update weight, then NPG is related to finding the minimizer for the following function:*

$$L(g; \theta, v) :=$$
$$\sum_{m=1}^{M} w_m \sum_{h=1}^{H} \mathbb{E}_{s, a \sim v_{m, H-h}} \left[ \left( A_{m,h}^{\pi_{\theta}}(s, a) - g^{\top} \nabla_{\theta} \ln \pi_{\theta}(a|s) \right)^2 \right].$$

**Definition 3** (Generic Fisher Information Matrix).

$$\Sigma_v^{\theta} := \sum_{m=1}^{M} w_m \sum_{h=1}^{H} \mathbb{E}_{s, a \sim v_{m, H-h}} \left[ (\nabla_{\theta} \ln \pi_{\theta}(a|s))^{\otimes} \right].$$

Particularly, denote $F(\theta) = \Sigma_{d^{\theta}}^{\theta}$ as the Fisher information matrix induced by $\pi_{\theta}$.

## 5 Learning Procedure

In this section we introduce the algorithms: NPG supporting any customized sampler, and our proposed Curriculum Learning framework.

**Natural Policy Gradient.** The learning procedure generates a series of parameters and policies. Starting from $\theta_0$, the algorithm updates the parameter by setting $\theta_{t+1} = \theta_t + \eta g_t$, where $\eta$ is a predefined constant learning rate, and $g_t$ is the update weight. Denote $\pi_t := \pi_{\theta_t}, V^t := V^{\pi_t}$ and $A_{m,h}^t := A_{m,h}^{\pi_t}$ for convenience. We adopt NPG (Kakade, 2002) because it is efficient in training parameterized policies and admits clean theoretical analysis. NPG satisfies $g_t \in \arg\min_g L(g; \theta_t, d^{\theta_t})$ (see Appendix D.1 for explanation). When we only have samples, we use the approximate version of NPG: $g_t \approx \arg\min_{g \in \mathcal{G}} L(g; \theta_t, d^{\theta_t})$, where $\mathcal{G} = \{x : \|x\|_2 \leqslant G\}$ for some hyper-parameter $G$.

We also introduce a variant of NPG: instead of sampling from $d^{\theta_t}$ using the current policy $\pi_t$, we sample from $\widetilde{d}^{\pi_s}$ using a *fixed* sampling policy $\pi_s$. The update rule is $g_t \approx \arg\min_{g \in \mathcal{G}} L(g; \theta_t, \widetilde{d}^{\pi_s})$. This version makes a closed-form analysis for BCP possible.

The main algorithm is shown in Algorithm 1. It admits two types of training: ① If $\pi_s = $ None, it calls Algorithm 4 (deferred to Appendix A) to sample $s, a \sim d^{\theta_t}$; ② If $\pi_s \neq$ None, it then calls Algorithm 4 to sample $s, a \sim \widetilde{d}^{\pi_s}$. Algorithm 4 also returns an unbiased estimation of $A_{H-h}^{\pi_t}(s, a)$.

In both cases, we denote $d^t$ as the sampling distribution and $\Sigma_t$ as the induced Fisher Information Matrix used in step $t$, i.e. $d^t := d^{\theta_t}, \Sigma_t := F(\theta_t)$ if $\pi_s = $ None; $d^t := \widetilde{d}^{\pi_s}, \Sigma_t := \Sigma_{\widetilde{d}^{\pi_s}}^{\theta_t}$ otherwise. The update rule can be written in a unified way as $g_t \approx \arg\min_{g \in \mathcal{G}} L(g; \theta_t, d^t)$. This is equivalent to solving a constrained quadratic optimization and we can use existing solvers.

**Remark 3.** Algorithm 1 is different from Algorithm 4 of Agarwal et al. (2021) in that we use a "batched" update while they used successive Projected Gradient Descents (PGD). This is an important implementation technique to speed up training in our experiments.

**Curriculum Learning.** We use Curriculum Learning to facilitate training. Algorithm 2 is our proposed training framework, which first constructs an easy environment $E'$ and trains a (near-)optimal policy $\pi_s$ of it. The design of $E'$ is problem-dependent. For the problems described in this paper (BCP, OKD, and ADW) as well as any similar problems (online load balancing, online set cover, etc.), we can use $n$, the sequence length of online decision-making, to represent the difficulty. For these problems, we construct $E'$ to be the environment with $n$ smaller than that of $E$. For other problems, we first find the hyperparameters controlling the difficulty of the problem, e.g., the sequence length, the action space size, the number of interaction steps, then reduce these hyperparameters to construct a smaller scale and simpler problem.

In the target environment $E$, we either use $\pi_s$ to sample data while training a new policy from scratch, or simply continue training $\pi_s$. To be specific and provide clarity for the results in Section 7, we name a few training modes (without regularization) here, and the rest are in Table 1 in Appendix C.

`curl`, the standard Curriculum Learning, runs Algorithm 2 with $samp = $ `pi_t`; `fix_samp_curl` stands for the fixed sampler Curriculum Learning, running Algorithm 2 with $samp = $ `pi_s`. `direct` means directly learning in $E$ without curriculum, i.e., running Algorithm 1 with $\pi_s = $ None; `naive_samp` also directly learns in $E$, while using $\pi_s = $ naïve random policy to sample data in Algorithm 1.

# 6 Performance Analysis

Our analysis contains two important components, namely the sub-optimality gap guarantee of the NPG we proposed, and the efficacy guarantee of Curriculum Learning on BCP. The first component can also be extended to history-dependent policies with features being the *tensor products* of features from each time step (exponentially large).

## 6.1 Natural Policy Gradient for Latent MDP

Let $g_t^\star \in \arg\min_{g \in \mathcal{G}} L(g; \theta_t, d^t)$ denote the true minimizer. We have the following definitions:

**Definition 4.** *Define for $0 \leqslant t \leqslant T$:*

- *(Excess risk)* $\epsilon_{\mathrm{stat}} := \max_t \mathbb{E}[L(g_t; \theta_t, d^t) - L(g_t^\star; \theta_t, d^t)]$;
- *(Transfer error)* $\epsilon_{\mathrm{bias}} := \max_t \mathbb{E}[L(g_t^\star; \theta_t, d^\star)]$;

---
**Algorithm 1** `NPG` (Full version: Algorithm 3)

---
1: **Input:** Environment $E$; learning rate $\eta$; episode number $T$; batch size $N$; initialization $\theta_0$; sampler $\pi_s$; optimization domain $\mathcal{G}$.
2: **for** $t \leftarrow 0, 1, \ldots, T-1$ **do**
3:     For $0 \leqslant n \leqslant N-1$ and $0 \leqslant h \leqslant H-1$, sample $(a_h^{(n)}, s_h^{(n)})$ and estimate $\widehat{A}_{H-h}^{(n)}$ using Algorithm 4.
4:     Calculate:

$$\widehat{F}_t \leftarrow \sum_{n=0}^{N-1} \sum_{h=0}^{H-1} (\nabla_\theta \ln \pi_{\theta_t}(a_h^{(n)}|s_h^{(n)}))^{\otimes},$$

$$\widehat{\nabla}_t \leftarrow \sum_{n=0}^{N-1} \sum_{h=0}^{H-1} \widehat{A}_{H-h}^{(n)} \nabla_\theta \ln \pi_{\theta_t}(a_h^{(n)}|s_h^{(n)}).$$

5:     Call any solver to get $\widehat{g}_t \leftarrow \arg\min_{g \in \mathcal{G}} g^\top \widehat{F}_t g - 2 g^\top \widehat{\nabla}_t$.
6:     Update $\theta_{t+1} \leftarrow \theta_t + \eta \widehat{g}_t$.
7: **end for**
8: **Return:** $\theta_T$.

---

---
**Algorithm 2** Curriculum learning framework.

---
1: **Input:** Environment $E$; learning rate $\eta$; episode number $T$; batch size $N$; sampler type $samp \in \{$ `pi_s`, `pi_t` $\}$; optimization domain $\mathcal{G}$.
2: Construct an environment $E'$ with a task easier than $E$. This environment should have optimal policy similar to that of $E$.
3: $\theta_s \leftarrow$ `NPG` $(E', \eta, T, N, 0^d, \text{None}, \mathcal{G})$ (Algorithm 1).
4: **if** $samp =$`pi_s` **then**
5:     $\theta_T \leftarrow$ `NPG` $(E, \eta, T, N, 0^d, \pi_s, \mathcal{G})$.
6: **else**
7:     $\theta_T \leftarrow$ `NPG` $(E, \eta, T, N, \theta_s, \text{None}, \mathcal{G})$.
8: **end if**
9: **Return:** $\theta_T$.

---

- *(Relative condition number)* $\kappa := \max_t \mathbb{E}\left[\sup_{x \in \mathbb{R}^d} \frac{x^\top \Sigma_{d^\star}^{\theta_t} x}{x^\top \Sigma_t x}\right]$. *Note that term inside the expectation is a random quantity as $\theta_t$ is random.*

*The expectation is with respect to the randomness in the sequence of weights $g_0, g_1, \ldots, g_T$.*

All the quantities are commonly used in literature mentioned in Section 2. $\epsilon_{\text{stat}}$ is due to that the minimizer $g_t$ from samples may not minimize the population loss $L$. $\epsilon_{\text{bias}}$ quantifies the approximation error due to feature maps. $\kappa$ characterizes the distribution mismatch between $d^t$ and $d^\star$ and is *a key quantity in Curriculum Learning* and will be studied in more details in the following sections.

Our main result is based on a fitting error which depicts the closeness between $\pi^\star$ and any policy $\pi$.

**Definition 5** (Fitting Error)**.** *Suppose the update rule is $\theta_{t+1} = \theta_t + \eta g_t$, define*

$$\text{err}_t := \sum_{m=1}^{M} w_m \sum_{h=1}^{H} \mathbb{E}_{(s,a) \sim d_{m,H-h}^\star}\left[A_{m,h}^t(s,a) - g_t^\top \nabla_\theta \ln \pi_t(a|s)\right].$$

Theorem 6 shows the convergence rate of Algorithm 1, and its proof is deferred to Appendix A.3.

**Theorem 6.** *With Definitions 4, 5 and 9, Algorithm 1 enjoys the following performance bound:*

$$\mathbb{E}\left[\min_{0 \leqslant t \leqslant T} V^\star - V^t\right] \leqslant \frac{\Phi(\pi_0)}{\eta T} + \eta \frac{B^2 G^2}{2} + \frac{1}{T} \sum_{t=0}^{T} \mathbb{E}[\text{err}_t]$$

$$\leqslant \frac{\Phi(\pi_0)}{\eta T} + \eta \frac{B^2 G^2}{2} + \sqrt{H \epsilon_{\text{bias}}} + \sqrt{H \kappa \epsilon_{\text{stat}}},$$

*where $\Phi(\pi_0)$ is the Lyapunov potential function which is only relevant to the initialization.*

**Remark 4.** ① For the results of sample-based NPG with entropy regularization for LMDP, please see Appendix A. ② Taking $\eta = \Theta(1/\sqrt{T})$ gives an $O(1/\sqrt{T})$ rate, matching the result in Agarwal et al. (2021). ③ $\epsilon_{\text{stat}}$ can be reduced using a larger batch size $N$ (Lemma 20) that $\epsilon_{\text{stat}} = \tilde{O}(1/\sqrt{N})$. ④ If some $d_t$ (especially the initialization $d_0$) is far away from $d^\star$, $\kappa$ may be extremely large (Section 6.2 as an example). If we can find a policy whose $\kappa$ is small with *a single curriculum*, we do not need the *multi-step curriculum* learning procedure used in Kong et al. (2019).

## 6.2 Curriculum learning for BCP

For BCP, there exists a threshold policy that is optimal (Beckmann, 1990). Suppose the threshold is $p \in (0, 1)$, then the policy is: accept the $i$-th number if and only if $i/n > p$ and $X_i = 1$. For the classical BCP where all the $n!$ instances have equal probability, the optimal threshold is $1/e$.

To show that curriculum learning makes the training converge faster, Theorem 6 gives a direct hint: *curriculum learning produces a good sampler leading to much smaller $\kappa$ than that of a naïve random sampler.* Here we focus on the cases where $samp = \texttt{pi\_s}$ because the sampler is fixed, while when $samp = \texttt{pi\_t}$ it is impossible to analyze a dynamic procedure. We show Theorem 7 to characterize $\kappa$ in BCP. Its full statement and proof is deferred to Appendix B.

**Theorem 7.** *Assume that each number is independent of others and the $i$-th number has a probability $P_i$ of being the maximum so far (Section 4.1). Assume the optimal policy is a $p$-threshold policy and the sampling policy is a $q$-threshold policy. There exists a policy parameterization such that:*

$$\kappa_{\text{curl}} = \Theta\left(\left\{ \begin{array}{ll} \prod_{j=\lfloor nq \rfloor+1}^{\lfloor np \rfloor} \frac{1}{1-P_j}, & q \leqslant p, \\ 1, & q > p, \end{array} \right.\right),$$

$$\kappa_{\text{naïve}} = \Theta\left( 2^{\lfloor np \rfloor} \max\left\{ 1, \max_{i \geqslant \lfloor np \rfloor+2} \prod_{j=\lfloor np \rfloor+1}^{i-1} 2(1-P_j) \right\} \right), \tag{1}$$

*where $\kappa_{\text{curl}}$ and $\kappa_{\text{naïve}}$ are $\kappa$ of the sampling policy and the naïve random policy, respectively.*

To understand how curriculum learning influences $\kappa$, we apply Theorem 7 to three concrete cases. They show that, when the state distribution induced by the optimal policy in the small problem is similar to that in the original large problem, then a single-step curriculum suffices (cf. ④ of Remark 4).

**The classical case: an exponential improvement.** We study the classical BCP first, where all the $n!$ permutations are sampled with equal probability. The probability series for this case is $P_i = 1/i$. Substituting them into Equation (1) directly gives:

$$\kappa_{\text{curl}} = \left\{ \begin{array}{ll} \frac{\lfloor n/e \rfloor}{\lfloor nq \rfloor}, & q \leqslant \frac{1}{e}, \\ 1, & q > \frac{1}{e}, \end{array} \right. \qquad \kappa_{\text{naïve}} = 2^{n-1} \frac{\lfloor n/e \rfloor}{n-1}.$$

Except for the corner case where $q < 1/n$, we have that $\kappa_{\text{curl}} = O(n)$ while $\kappa_{\text{naïve}} = \Omega(2^n)$. Notice that any distribution with $P_i \leqslant 1/i$ leads to an exponential improvement.

**A more general case.** Now we try to loosen the condition where $P_i \leqslant 1/i$. Let us consider the case where $P_i \leqslant 1/2$ for $i \geqslant 2$ (by definition $P_1$ is always equal to 1). Equation (1) now becomes:

$$\kappa_{\text{curl}} \leqslant \left\{ \begin{array}{ll} 2^{\lfloor np \rfloor - \lfloor nq \rfloor}, & q \leqslant p, \\ 1, & q > p, \end{array} \right. \qquad \kappa_{\text{naïve}} \geqslant 2^{\lfloor np \rfloor}.$$

Clearly, $\kappa_{\text{curl}} \leqslant \kappa_{\text{naïve}}$ always holds. When $q$ is close to $p$, the difference is exponential in $\lfloor nq \rfloor$.

**Failure mode of Curriculum Learning.** Lastly we show further relaxing the assumption on $P_i$ leads to failure cases. The extreme case is that all $P_i = 1$, i.e., the maximum number always comes as the last one. Suppose $q < 1 - 1/n$, then $d^{\pi_q}(1) = 0$. Hence $\kappa_{\text{curl}} = \infty$, larger than $\kappa_{\text{naïve}} = 2^{n-1}$. From Equation (1), $\kappa_{\text{naïve}} \leqslant 2^{n-1}$. Similar as Section 3 of Beckmann (1990), the optimal threshold $p$ satisfies:

$$\sum_{i=\lfloor np \rfloor+2}^{n} \frac{P_i}{1-P_i} \leqslant 1 < \sum_{i=\lfloor np \rfloor+1}^{n} \frac{P_i}{1-P_i}.$$

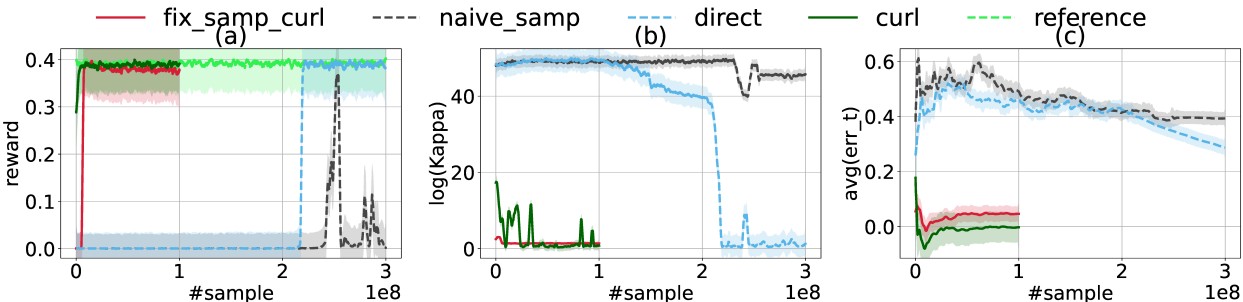

Figure 1: One experiment of BCP. The $x$-axis is the number of trajectories, i.e., number of episodes $\times$ horizon $\times$ batch size. **Dashed lines represent only final phase training and solid lines represent Curriculum Learning.** The shadowed area shows the 95% confidence interval for the expectation. The explanation for different modes can be found in Section 5. The reference policy is the optimal threshold policy.

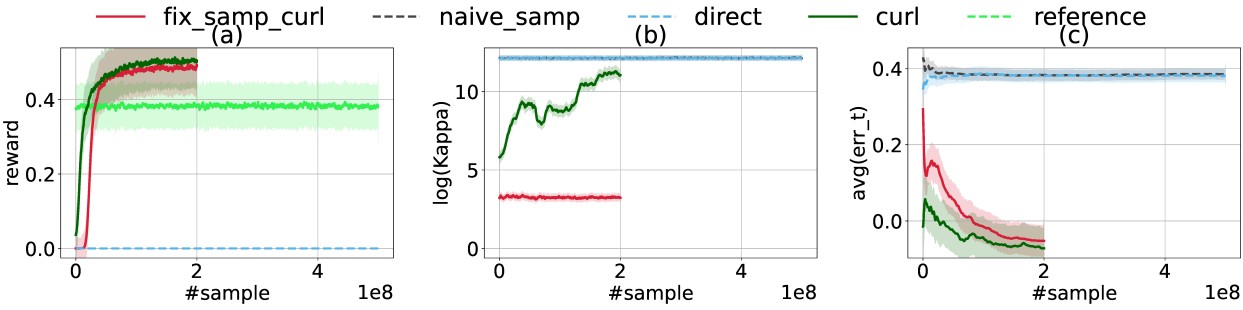

Figure 2: One experiment of OKD. Legend description is the same as that of Figure 1. The reference policy is the bang-per-buck algorithm for Online Knapsack (Section 3.1 of Kong et al. (2019)).

So letting $P_n > 1/2$ results in $p \in [1 - 1/n, 1)$. Further, if $q < 1 - 1/n$ and $P_j > 1 - 2^{-\frac{n}{n-\lfloor nq \rfloor - 1}}$ for any $\lfloor nq \rfloor + 1 \leqslant j \leqslant n - 1$, then from Equation (1), $\kappa_{\text{curl}} > 2^n > \kappa_{\text{naïve}}$. This means that Curriculum Learning can always be manipulated adversarially. Sometimes there is hardly any reasonable curriculum.

**Remark 5.** Here we only provide theoretical explanations for BCP when $samp = \texttt{pi\_s}$, because $\kappa$ is highly problem-dependent, and the analytical forms for $\kappa$ is tractable when the sampler is fixed. For $samp = \texttt{pi\_t}$ and other CO problems such as OKD, however, we do not have analytical forms, so we resort to empirical studies (Section 7).

## 7 Experiments

The experiments' formulations are modified from Kong et al. (2019). Due to page limit, more formulation details and results are presented in Appendix C, and code can be found at `https://github.com/zhourunlong/RL-for-Combinatorial-Optimization`. In Curriculum Learning the entire training process splits into at most two phases. We call the training on curriculum (small scale instances) "warm-up phase" and the training on large scale instances "final phase". If the training is directly on large scale instances, we still call it "final phase" for convenience. For each problem, we run multiple experiments using different distributions of instances. Each experiment contains multiple training methods, e.g., direct training, curriculum learning, etc. To highlight the effect of curriculum learning, we omit the results regarding regularization, and they can be found in supplementary files. All the trainings in the same experiment have *the same distributions* over LMDPs for final phase and warm-up phase (if any), respectively.

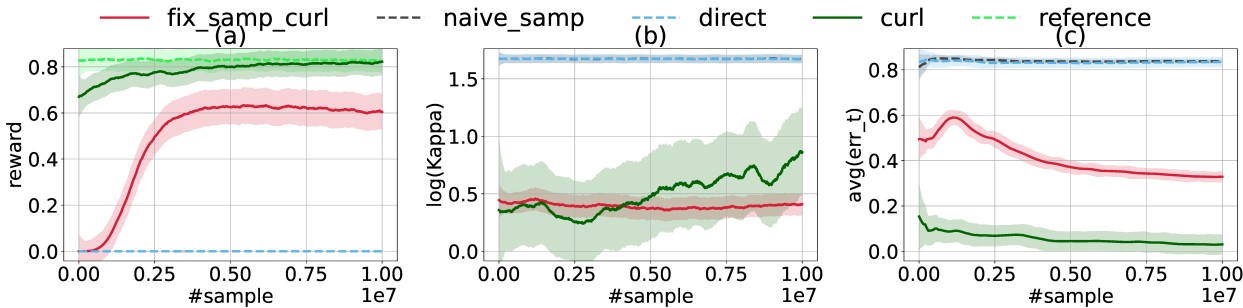

Figure 3: One experiment of ADW. Legend description is the same as that of Figure 1. The reference policy is obtained by running a `curl` procedure. The $\ln\kappa$ and $\mathrm{avg}(\mathrm{err}_t)$ curves are then plotted with the above reference policy hard-coded into the environment.

**The Best Choice Problem (BCP).** We show one of the four experiments in Figure 1. Aside from reward and $\ln\kappa$, we plot the weighted average of $\mathrm{err}_t$ according to Theorem 6: $\mathrm{avg}(\mathrm{err}_t) = \sum_{i=0}^{t} \mathrm{err}_i/T$. All the instance distributions are generated from parameterized series $\{P_n\}$ with fixed random seeds, which guarantees reproducibility and comparability. Aside from the fact that *the curriculum is a smaller BCP*, there is *no other explicit relationship between the curriculum and the target environment*, so the curriculum can be viewed as *random and independent*. The experiments clearly demonstrate that *curriculum learning can boost the performance by a large margin* and curriculum learning indeed dramatically reduces $\kappa$, even the curriculum is randomly generated.

**Online Knapsack (decision version, OKD).** We show one of the three experiments in Figure 2. $\ln\kappa$ and $\mathrm{avg}(\mathrm{err}_t)$ are with respect to the reference policy, a bang-per-buck algorithm, which is not the optimal policy. Thus, they are only for reference. The curriculum generation is also parameterized, random and independent of the target environment. The experiments again demonstrate the effectiveness of curriculum learning and curriculum learning indeed dramatically reduces $\kappa$.

**AdWords (decision version, ADW).** We show one of the two experiments in Figure 3. The reference policy is obtained by using curriculum learning and training until nearly convergence. The curriculum generation is also parameterized, random and independent of the target environment. The experiments again demonstrate the effectiveness of curriculum learning.

## 8    Conclusion

We showed online CO problems could be naturally formulated as LMDPs, and we analyzed the convergence rate of NPG for LMDPs. Our theory shows the main benefit of curriculum learning is finding a stronger sampling strategy, especially for classical BCP any curriculum exponentially improves the learning rate. Our empirical results on BCP, OKD, and ADW also corroborated our findings. Our work is the first attempt to systematically study techniques devoted to using RL to tackle online CO problems, which we believe is a fruitful direction worth further investigations.

## Acknowledgement

SSD acknowledges the support of NSF IIS 2110170, NSF DMS 2134106, NSF CCF 2212261, NSF IIS 2143493, NSF CCF 2019844, NSF IIS 2229881.

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

# A  Full Results of the Main Algorithm and Theorem for Entropy Regularization

## A.1  Notations and Definitions

**Entropy regularized value function, Q-function and advantage function.**  We incorporate *entropy regularization* for completeness because prior works (especially empirical works) used it to facilitate training. We define the value function in a unified way: $V_{m,h}^{\pi,\lambda}(s)$ is defined as the sum of future $\lambda$-regularized rewards starting from $s$ and executing $\pi$ for $h$ steps in $M_m$, i.e.,

$$V_{m,h}^{\pi,\lambda}(s) := \mathbb{E}_{\mathcal{M}_m,\pi}\left[\sum_{t=0}^{h-1} r_m^{\pi,\lambda}(s_t, a_t) \;\middle|\; s_0 = s\right],$$

where $r_m^{\pi,\lambda}(s,a) := r_m(s,a) + \lambda \ln \frac{1}{\pi(a|s)}$, and the expectation is with respect to the randomness of trajectory induced by $\pi$ in $M_m$. Clearly, $V_{m,h}^{\pi}(s) = V_{m,h}^{\pi,0}(s)$.

For any $\mathcal{M}_m, \pi, h$, with $\mathcal{H}(\pi(\cdot|s)) := \sum_{a\in\mathcal{A}} \pi(a|s) \ln \frac{1}{\pi(a|s)} \in [0, \ln|\mathcal{A}|]$ we define

$$H_{m,h}^{\pi}(s) := \mathbb{E}_{\mathcal{M}_m,\pi}\left[\sum_{t=0}^{h-1} \mathcal{H}(\pi(\cdot|s_t)) \;\middle|\; s_0 = s\right].$$

In fact, $V_{m,h}^{\pi,\lambda}(s) = V_{m,h}^{\pi}(s) + \lambda H_{m,h}^{\pi}(s)$.

Denote $V^{\pi,\lambda} := \sum_{m=1}^{M} w_m \sum_{s_0\in\mathcal{S}} \nu_m(s_0) V_{m,H}^{\pi,\lambda}(s_0)$ then $V^{\pi} = V^{\pi,0}$. The original goal is to find $\pi^{\star} = \arg\max_{\pi\in\Pi} V^{\pi}$. Under regularization, we seek for $\pi_\lambda^{\star} = \arg\max_{\pi\in\Pi} V^{\pi,\lambda}$ instead. Denote $V^{\star,\lambda} = V^{\pi_\lambda^{\star},\lambda}$.

Since $V^\star \leqslant V^{\pi^\star,\lambda} \leqslant V^{\star,\lambda} \leqslant V^{\pi_\lambda^\star} + \lambda H \ln |\mathcal{A}|$, the regularized optimal policy $\pi_\lambda^\star$ can be nearly optimal as long as the regularization coefficient $\lambda$ is small enough. For notational ease, we abuse $\pi^\star$ with $\pi_\lambda^\star$.

The Q-function can be defined in a similar manner:

$$Q_{m,h}^{\pi,\lambda}(s,a) := \mathbb{E}_{\mathcal{M}_m,\pi} \left[ \sum_{t=0}^{h-1} r_m^{\pi,\lambda}(s_t, a_t) \;\middle|\; (s_0, a_0) = (s,a) \right],$$

and the advantage function is defined as $A_{m,h}^{\pi,\lambda}(s,a) := Q_{m,h}^{\pi,\lambda}(s,a) - V_{m,h}^{\pi,\lambda}(s)$.

Denote $\pi_t := \pi_{\theta_t}, V^{t,\lambda} := V^{\pi_t,\lambda}$ and $A_{m,h}^{t,\lambda} := A_{m,h}^{\pi_t,\lambda}$ for convenience.

**Definition 8** (Definition 2 with entropy regularization)**.** *Let $g$ be the parameter update weight, then NPG is related to finding the minimizer for the following function:*

$$L(g; \theta, v) := \sum_{m=1}^{M} w_m \sum_{h=1}^{H} \mathbb{E}_{s,a \sim v_{m,H-h}} \left[ \left( A_{m,h}^{\pi_\theta,\lambda}(s,a) - g^\top \nabla_\theta \ln \pi_\theta(a|s) \right)^2 \right].$$

**Definition 9** (Lyapunov Potential Function (Cayci et al., 2021))**.** *We define the potential function $\Phi : \Pi \to \mathbb{R}$ as follows: for any $\pi \in \Pi$,*

$$\Phi(\pi) = \sum_{m=1}^{M} w_m \sum_{h=0}^{H-1} \mathbb{E}_{(s,a) \sim d_{m,h}^\star} \left[ \ln \frac{\pi^\star(a|s)}{\pi(a|s)} \right].$$

## A.2 Algorithms

Algorithm 3 is the full version of Algorithm 1, with support of entropy regularization. Algorithm 4 is the skipped sampling function.

## A.3 Performance of Natural Policy Gradient for LMDP

We restate Theorem 6 with entropy regularization.

**Theorem 6** (Full Statement of Theorem 6)**.** *With Definitions 4, 5 and 9, Algorithm 3 enjoys the following performance bound:*

$$\mathbb{E}\left[ \min_{0 \leqslant t \leqslant T} V^{\star,\lambda} - V^{t,\lambda} \right] \leqslant \frac{\lambda(1-\eta\lambda)^{T+1}\Phi(\pi_0)}{1-(1-\eta\lambda)^{T+1}} + \eta \frac{B^2 G^2}{2} + \frac{\sum_{t=0}^{T}(1-\eta\lambda)^{T-t}\mathbb{E}[\mathrm{err}_t]}{\sum_{t'=0}^{T}(1-\eta\lambda)^{T-t'}}$$

$$\leqslant \frac{\lambda(1-\eta\lambda)^{T+1}\Phi(\pi_0)}{1-(1-\eta\lambda)^{T+1}} + \eta \frac{B^2 G^2}{2} + \sqrt{H\epsilon_{\mathrm{bias}}} + \sqrt{H\kappa\epsilon_{\mathrm{stat}}}.$$

*Proof.* Here we make shorthands for the sub-optimality gap and potential function: $\Delta_t := V^{\star,\lambda} - V^{t,\lambda}$ and $\Phi_t := \Phi(\pi_t)$. From Lemma 16 we have

$$\eta\Delta_t \leqslant (1-\eta\lambda)\Phi_t - \Phi_{t+1} + \eta\mathrm{err}_t + \eta^2 \frac{B^2 G^2}{2}.$$

Taking expectation over the update weights, we have

$$\mathbb{E}[\eta\Delta_t] \leqslant (1-\eta\lambda)\mathbb{E}[\Phi_t] - \mathbb{E}[\Phi_{t+1}] + \eta\mathbb{E}[\mathrm{err}_t] + \eta^2 \frac{B^2 G^2}{2}.$$

Thus,

$$\mathbb{E}\left[ \eta \sum_{t=0}^{T}(1-\eta\lambda)^{T-t}\Delta_t \right] \leqslant \sum_{t=0}^{T}(1-\eta\lambda)^{T-t+1}\mathbb{E}[\Phi_t] - \sum_{t=0}^{T}(1-\eta\lambda)^{T-t}\mathbb{E}[\Phi_{t+1}]$$

---

**Algorithm 3** `NPG`: Sample-based NPG (full version).

---

1: **Input:** Environment $E$; learning rate $\eta$; episode number $T$; batch size $N$; initialization $\theta_0$; sampler $\pi_s$; regularization coefficient $\lambda$; entropy clip bound $U$; optimization domain $\mathcal{G}$.

2: **for** $t \leftarrow 0, 1, \ldots, T-1$ **do**

3:     Initialize $\widehat{F}_t \leftarrow 0^{d \times d}, \widehat{\nabla}_t \leftarrow 0^d$.

4:     **for** $n \leftarrow 0, 1, \ldots, N-1$ **do**

5:         **for** $h \leftarrow 0, 1, \ldots, H-1$ **do**

6:             **if** $\pi_s$ is not None **then**

7:                 $s_h, a_h, \widehat{A}_{H-h}(s_h, a_h) \leftarrow$ Sample $(E, \pi_s, \text{True}, \pi_t, h, \lambda, U)$ (see Algorithm 4).

                // $s, a \sim \widetilde{d}_{m,h}^{\pi_s}$, estimate $A_{m,H-h}^{t,\lambda}(s, a)$.

8:             **else**

9:                 $s_h, a_h, \widehat{A}_{H-h}(s_h, a_h) \leftarrow$ Sample $(E, \pi_t, \text{False}, \pi_t, h, \lambda, U)$.

                // $s, a \sim d_{m,h}^{\theta_t}$, estimate $A_{m,H-h}^{t,\lambda}(s, a)$.

10:             **end if**

11:         **end for**

12:         Update:

$$\widehat{F}_t \leftarrow \widehat{F}_t + \sum_{h=0}^{H-1} \nabla_\theta \ln \pi_{\theta_t}(a_h | s_h) \left( \nabla_\theta \ln \pi_{\theta_t}(a_h | s_h) \right)^\top,$$

$$\widehat{\nabla}_t \leftarrow \widehat{\nabla}_t + \sum_{h=0}^{H-1} \widehat{A}_{H-h}(s_h, a_h) \nabla_\theta \ln \pi_{\theta_t}(a_h | s_h).$$

13:     **end for**

14:     Call any solver to get $\widehat{g}_t \leftarrow \arg\min_{g \in \mathcal{G}} g^\top \widehat{F}_t g - 2g^\top \widehat{\nabla}_t$.

15:     Update $\theta_{t+1} \leftarrow \theta_t + \eta \widehat{g}_t$.

16: **end for**

17: **Return:** $\theta_T$.

---

$$+ \eta \sum_{t=0}^{T} (1 - \eta\lambda)^{T-t} \mathbb{E}[\text{err}_t] + \eta^2 \frac{B^2 G^2}{2} \sum_{t=0}^{T} (1 - \eta\lambda)^{T-t}$$

$$= (1 - \eta\lambda)^{T+1} \Phi_0 - \mathbb{E}[\Phi_{T+1}] + \eta \sum_{t=0}^{T} (1 - \eta\lambda)^{T-t} \mathbb{E}[\text{err}_t] + \eta^2 \frac{B^2 G^2}{2} \sum_{t=0}^{T} (1 - \eta\lambda)^{T-t}$$

$$\leqslant (1 - \eta\lambda)^{T+1} \Phi_0 + \eta \sum_{t=0}^{T} (1 - \eta\lambda)^{T-t} \mathbb{E}[\text{err}_t] + \eta^2 \frac{B^2 G^2}{2} \sum_{t=0}^{T} (1 - \eta\lambda)^{T-t},$$

where the last step uses the fact that $\Phi(\pi) \geqslant 0$. This is a weighted average, so by normalizing the coefficients,

$$\mathbb{E}\left[ \min_{0 \leqslant t \leqslant T} \Delta_t \right] \leqslant \frac{\lambda(1 - \eta\lambda)^{T+1} \Phi_0}{1 - (1 - \eta\lambda)^{T+1}} + \eta \frac{B^2 G^2}{2} + \frac{\sum_{t=0}^{T} (1 - \eta\lambda)^{T-t} \mathbb{E}[\text{err}_t]}{\sum_{t'=0}^{T} (1 - \eta\lambda)^{T-t'}}$$

$$\leqslant \frac{\lambda(1 - \eta\lambda)^{T+1} \Phi_0}{1 - (1 - \eta\lambda)^{T+1}} + \eta \frac{B^2 G^2}{2} + \sqrt{H \epsilon_{\text{bias}}} + \sqrt{H \kappa \epsilon_{\text{stat}}},$$

where the last step comes from Lemma 17 and Jensen's inequality. This completes the proof. $\qquad \square$

Aside from Remark 4, we have extra remarks:

**Remark 6.** ① This is the first result for LMDP and sample-based NPG with entropy regularization. ② For any fixed $\lambda > 0$ we have a linear convergence, which matches the result of discounted infinite horizon MDP (Theorem 1 in Cayci et al. (2021)); the limit when $\lambda$ tends to 0 is $O(1/(\eta T) + \eta)$ (which implies an $O(1/\sqrt{T})$ rate), matching the result in Agarwal et al. (2021).

---

**Algorithm 4** Sample: Sampler for $s \sim d_{m,h}^{\pi_{\mathrm{samp}}}$ where $m \sim$ Multinomial $(w_1, \ldots, w_M)$, $a \sim \mathrm{Unif}_{\mathcal{A}}$ if $unif =$ True and $a \sim \pi_{\mathrm{samp}}(\cdot|s)$ otherwise, and estimate of $A_{m,H-h}^{t,\lambda}(s,a)$.

---

1: **Input:** Environment $E$; sampler policy $\pi_{\mathrm{samp}}$; whether to sample uniform actions after state $unif$; current policy $\pi_t$; time step $h$; regularization coefficient $\lambda$; entropy clip bound $U$.
2: $E.\mathrm{reset}()$.
3: **for** $i \leftarrow 0, 1, \ldots, h-1$ **do**
4:     $s_i \leftarrow E.\mathrm{get\_state}()$.
5:     Sample action $a_i \sim \pi_{\mathrm{samp}}(\cdot|s_i)$ and $E.\mathrm{execute}(a_i)$.
6: **end for**
7: $s_h \leftarrow E.\mathrm{get\_state}()$.
8: **if** $unif =$ True **then**
9:     $a_h \sim \mathrm{Unif}_{\mathcal{A}}$.
10: **else**
11:     $a_h \sim \pi_{\mathrm{samp}}(\cdot|s_h)$.
12: **end if**
13: $(s,a) \leftarrow (s_h, a_h)$.
14: Get a random number $p \sim \mathrm{Unif}[0,1]$.
15: **if** $p < \frac{1}{2}$ **then**
16:     Override $a_h \sim \pi_t(\cdot|s_h)$.
17:     Set importance weight $C \leftarrow -2$.
18:     $r_h \leftarrow E.\mathrm{execute}(a_h)$.
19:     Initialize cumulative reward $R \leftarrow r_h + \lambda \mathcal{H}(\pi_t(\cdot|s_h))$.
20: **else**
21:     $C \leftarrow 2$.
22:     $r_h \leftarrow E.\mathrm{execute}(a_h)$.
23:     $R \leftarrow r_h + \lambda \min\{\ln \frac{1}{\pi_t(a_h|s_h)}, U\}$.
24: **end if**
25: **for** $i \leftarrow h+1, h+2, \ldots, H-1$ **do**
26:     $s_i \leftarrow E.\mathrm{get\_state}()$.
27:     $a_i \sim \pi_t(\cdot|s_i)$ and $r_h \leftarrow E.\mathrm{execute}(a_i)$.
28:     $R \leftarrow R + r_i + \lambda \mathcal{H}(\pi_t(\cdot|s_i))$.
29: **end for**
30: **Return:** $s, a, \widehat{A}_{H-h}^{t,\lambda}(s,a) = CR$.

---

## B    Results of Curriculum Learning for the Best Choice Problem (BCP)

**Theorem 7** (Formal statement of Theorem 7). *For BCP, set $samp = $ `pi_s` in Algorithm 2. Assume that each number is independent from others and the $i$-th number has probability $P_i$ of being the best so far (see formulation in Section 4.1 and Appendix C.1). Assume the optimal policy is a $p$-threshold policy and the sampling policy is a $q$-threshold policy. There exists a policy parameterization and quantities*

$$
k_{\mathrm{curl}} = \begin{cases} \prod_{j=\lfloor nq \rfloor+1}^{\lfloor np \rfloor} \frac{1}{1-P_j}, & q \leqslant p, \\ 1, & q > p, \end{cases} \qquad k_{\mathrm{naïve}} = 2^{\lfloor np \rfloor} \max \left\{ 1, \max_{i \geqslant \lfloor np \rfloor+2} \prod_{j=\lfloor np \rfloor+1}^{i-1} 2(1-P_j) \right\},
$$

*such that $k_{\mathrm{curl}} \leqslant \kappa_{\mathrm{curl}} \leqslant 2k_{\mathrm{curl}}$ and $k_{\mathrm{naïve}} \leqslant \kappa_{\mathrm{naïve}} \leqslant 2k_{\mathrm{naïve}}$. Here $\kappa_{\mathrm{curl}}$ and $\kappa_{\mathrm{naïve}}$ correspond to $\kappa$ induced by the $q$-threshold policy and the naïve random policy respectively.*

*Proof.* We need to calculate three state-action visitation distributions: that induced by the optimal policy, $d^\star$; that induced by the sampler which is the optimal for the curriculum, $\widetilde{d}^{\mathrm{curl}}$; and that induced by the naïve random sampler, $\widetilde{d}^{\mathrm{naïve}}$. This then boils down to calculating the state(-action) visitation distribution under two types of policies: any threshold policy and the naïve random policy.

For any policy $\pi$, denote $d^\pi(i/n)$ as the probability for the agent acting under $\pi$ to see states $i/n$ with arbitrary $x_i$. We do not need to take the terminal state $g$ into consideration, since it stays in a zero-reward loop and contributes 0 to $L(g; \theta, d)$. We use the LMDP distribution parameterization $\{P_n\}$ described in Section 7.

Denote $\pi_p$ as the $p$-threshold policy, i.e. accept if and only if $i/n > p$ and $x_i = 1$. Then

$$d^{\pi_p}\left(\frac{i}{n}\right) = \mathbb{P}(\text{reject all previous } i-1 \text{ states}|\pi_p)$$

$$= \prod_{j=1}^{i-1}\left(\mathbb{P}\left(\frac{j}{n}, 1\right)\mathbb{1}\left[\frac{j}{n} \leqslant p\right] + 1 - \mathbb{P}\left(\frac{j}{n}, 1\right)\right)$$

$$= \prod_{j=\lfloor np \rfloor + 1}^{i-1}\left(1 - \mathbb{P}\left(\frac{j}{n}, 1\right)\right)$$

$$= \prod_{j=\lfloor np \rfloor + 1}^{i-1}(1 - P_j).$$

Denote $\pi_{\text{naïve}}$ as the naïve random policy, i.e., accept any number with probability $1/2$ regardless of the state. Then

$$d^{\pi_{\text{naïve}}}\left(\frac{i}{n}\right) = \mathbb{P}(\text{reject all previous } i-1 \text{ states}|\pi_{\text{naïve}}) = \frac{1}{2^{i-1}}.$$

For any $\pi$, we can see that the state visitation distribution satisfies $d^\pi(i/n, 1) = P_i d^\pi(i/n)$ and $d^\pi(i/n, 0) = (1 - P_i)d^\pi(i/n)$.

To show the possible largest difference, we use a parameterization that for each state $s$, $\phi(s) = \text{One-hot}(s)$. The policy is then satisfied into

$$\pi_\theta(\text{accept}|s) = \frac{\exp(\theta^\top \phi(s))}{\exp(\theta^\top \phi(s)) + 1}, \quad \pi_\theta(\text{reject}|s) = \frac{1}{\exp(\theta^\top \phi(s)) + 1},$$

because there are only two actions. Denote $\pi_\theta(s) = \pi_\theta(\text{accept}|s)$, we have

$$\nabla_\theta \ln \pi_\theta(\text{accept}|s) = (1 - \pi_\theta(s))\phi(s), \quad \nabla_\theta \ln \pi_\theta(\text{reject}|s) = -\pi_\theta(s)\phi(s).$$

Now suppose the optimal threshold and the threshold learned through curriculum are $p$ and $q$, then

$$\Sigma_{d^\star}^\theta = \sum_{s \in \mathcal{S}} d^{\pi_p}(s)\left(\pi_p(s)(1 - \pi_\theta(s))^2 + (1 - \pi_p(s))\pi_\theta(s)^2\right)\phi(s)\phi(s)^\top,$$

$$\Sigma_{\tilde{d}^{\text{curl}}}^\theta = \sum_{s \in \mathcal{S}} d^{\pi_q}(s)\left(\frac{1}{2}(1 - \pi_\theta(s))^2 + \frac{1}{2}\pi_\theta(s)^2\right)\phi(s)\phi(s)^\top,$$

$$\Sigma_{\tilde{d}^{\text{naïve}}}^\theta = \sum_{s \in \mathcal{S}} d^{\text{naïve}}(s)\left(\frac{1}{2}(1 - \pi_\theta(s))^2 + \frac{1}{2}\pi_\theta(s)^2\right)\phi(s)\phi(s)^\top.$$

Denote $\kappa_\clubsuit(\theta) = \sup_{x \in \mathbb{R}^d} \frac{x^\top \Sigma_{d^\star}^\theta x}{x^\top \Sigma_{d^\clubsuit}^\theta x}$. From parameterization we know all $\phi(s)$ are orthogonal. Abusing $\pi_q$ with $\pi_{\text{curl}}$, we have

$$\kappa_\clubsuit(\theta) = \max_{s \in \mathcal{S}} \frac{d^{\pi_p}(s)\left(\pi^\star(s)(1 - \pi_\theta(s))^2 + (1 - \pi^\star(s))\pi_\theta(s)^2\right)}{d^\clubsuit(s)\left(\frac{1}{2}(1 - \pi_\theta(s))^2 + \frac{1}{2}\pi_\theta(s)^2\right)}.$$

We can separately consider each $s \in \mathcal{S}$ because of the orthogonal features. Observe that $\pi_p(s) \in \{0, 1\}$, so for $s \in \mathcal{S}$, its corresponding term in $\kappa_\clubsuit(\theta)$ is maximized when $\pi_\theta(s) = 1 - \pi_p(s)$ and is equal to $2\frac{d^{\pi_p}(s)}{d^\clubsuit(s)}$.

By definition, $\kappa_{\clubsuit} = \max_{0 \leqslant t \leqslant T} \mathbb{E}[\kappa_{\clubsuit}(\theta_t)]$. Since $\theta_0 = 0^d$, we have $\kappa_{\clubsuit} \geqslant \kappa_{\clubsuit}(0^d)$ where $\pi_\theta(s) = 1/2$ and the corresponding term is $\frac{d^{\pi_p}(s)}{d^{\clubsuit}(s)}$. So

$$\max_{s \in \mathcal{S}} \frac{d^{\pi_p}(s)}{d^{\clubsuit}(s)} \leqslant \kappa_{\clubsuit} \leqslant 2 \max_{s \in \mathcal{S}} \frac{d^{\pi_p}(s)}{d^{\clubsuit}(s)}.$$

We now have an order-accurate result $k_{\clubsuit} = \max_{s \in \mathcal{S}} \frac{d^{\pi_p}(s)}{d^{\clubsuit}(s)}$ for $\kappa_{\clubsuit}$. Direct computation gives

$$k_{\text{curl}} = \begin{cases} \prod_{j=\lfloor nq \rfloor+1}^{\lfloor np \rfloor} \frac{1}{1-P_j}, & q \leqslant p, \\ 1, & q > p, \end{cases}$$

$$k_{\text{naïve}} = 2^{\lfloor np \rfloor} \max \left\{ 1, \max_{i \geqslant \lfloor np \rfloor+2} \prod_{j=\lfloor np \rfloor+1}^{i-1} 2(1-P_j) \right\}.$$

This completes the proof. $\qquad\qquad\qquad\qquad\qquad\qquad\qquad\qquad\qquad\qquad\qquad\qquad\qquad\qquad\square$

## C   Full Experiments

Here are all the experiments not shown in Section 7. All the experiments were run on a server with CPU AMD Ryzen 9 3950X, GPU NVIDIA GeForce 2080 Super and 128G memory. For legend description please refer to the caption of Figure 1. For code please refer to `https://github.com/zhourunlong/RL-for-Combinatorial-Optimization`.

**Policy parameterization.**

- For BCP and OKD, there are exactly two actions, so we can use $\phi(s) = \phi(s, \text{accept}) - \phi(s, \text{reject})$ instead of $\phi(s, \text{accept})$ and $\phi(s, \text{reject})$. Now the policy is $\pi_\theta(\text{accept}|s) = \frac{\exp(\theta^\top \phi(s))}{\exp(\theta^\top \phi(s))+1}$ and $\pi_\theta(\text{reject}|s) = \frac{1}{\exp(\theta^\top \phi(s))+1}$.

- For ADW, there are $n+1$ actions ($n$ for assigning a slot to advertisers and 1 for not assigning it). So, we must follow the canonical form of log-linear policies.

**Training schemes.** We ran nine experiments in total, four for BCP, three for OKD, and two for ADW. The difference between the experiments of the same problem lies in the distribution over instances (i.e., $\{w_m\}$). In the following subsections, we will introduce how we parameterized the distribution in detail. In a single experiment, we ran eight setups, each representing a combination of sampler policies, initialization policies of the final phase, and whether we used regularization. For visual clarity, we did not plot setups with entropy regularization, but the readers can plot it using `plot.py` in the supplementary files. We make a detailed list of the training schemes in Table 1.

### C.1   The Best Choice Problem (BCP)

**State and action spaces.** States with $X_i > 1$ are the same. To make the problem "scale-invariant", we use $i/n$ to represent $i$. So the states are $s = (i/n, x_i = \mathbb{1}[X_i = 1])$. There is an additional terminal state $g = (0, 0)$. For each state, the agent can either accept or reject.

**Transition and reward.** Any action in $g$ leads back to $g$. Once the agent accepts the $i$-th number, the state transits into $g$, and reward is 1 if $i$ is the maximum in the instance. If the agent rejects, then the state goes to $((i+1)/n, x_{i+1})$ if $i < n$ and $g$ if $i = n$. For all other cases, rewards are 0.

**Feature mapping.** Recall that all states are of the form $(f, x)$ where $f \in [0, 1]$, $x \in \{0, 1\}$. We set a degree $d_0$ and the feature mapping is constructed as the collection of polynomial bases with degree less than $d_0$ ($d = 2d_0$):

$$\phi(f, x) = (1, f, \ldots, f^{d_0-1}, x, fx, \ldots, f^{d_0-1}x).$$

| Abbreviation | Detailed setup | Script |
|---|---|---|
| `fix_samp_curl` | **Fix**ed **samp**ler **cur**riculum learning. In the warm-up phase, train a policy $\pi_s$ from scratch (with zero initialization in parameters) using a small environment $E'$. In the final phase, change to the true environment $E$, use $\pi_s$ as the sampler policy to train a policy from scratch. | Run Alg. 2 with $samp = $ `pi_s` and $\lambda = 0$. |
| `fix_samp_curl_reg` | The same as `fix_samp_curl`, but add entropy **reg**ularization to both phases. | Run Alg. 2 with $samp = $ `pi_s` and $\lambda \neq 0$. |
| `direct` | **Direct** learning. Only the final phase. Train a policy from scratch directly in $E$. | Run Alg. 1 with $\theta_0 = 0^d$, $\pi_s = $ None and $\lambda = 0$. |
| `direct_reg` | The same as `direct`, but add entropy **reg**ularization. | Run Alg. 1 with $\theta_0 = 0^d$, $\pi_s = $ None and $\lambda \neq 0$. |
| `naive_samp` | Learning with the **naïve samp**ler. Only the final phase. Use the naïve random policy as the sampler to train a policy from scratch in $E$. | Run Alg. 1 with $\theta_0 = 0^d$, $\pi_s = $ naïve random policy and $\lambda = 0$. |
| `naive_samp_reg` | The same as `naive_samp`, but add entropy **reg**ularization. | Run Alg. 1 with $\theta_0 = 0^d$, $\pi_s = $ naive random policy and $\lambda \neq 0$. |
| `curl` | **Cur**riculum **l**earning. In the warm-up phase, train a policy $\pi_s$ from scratch in $E'$. In the final phase, change to $E$ and continue on training $\pi_s$. | Run Alg. 2 with $samp = $ `pi_t` and $\lambda = 0$. |
| `curl_reg` | The same as `curl`, but add entropy **reg**ularization. | Run Alg. 2 with $samp = $ `pi_t` and $\lambda \neq 0$. |
| `reference` | This is the **reference** policy. For BCP, it is exactly the optimal policy since it can be calculated. For OKD, it is a bang-per-buck policy and is not the optimal policy (whose exact form is not clear). For ADW, it is the near optimal policy in our restricted policy / feature class (trained using curriculum learning). | N/A |

Table 1: Detailed setups for each training scheme.

**LMDP distribution.** We model the distribution as follows: for each $i$, we can have $x_i = 1$ with probability $P_i$ and is independent from other $i'$. By definition, $P_1 = 1$ while other $P_i$ can be arbitrary. The classical BCP satisfies $P_i = 1/i$. We also experimented on three other distributions (so in total there are four experiments), each with a series of numbers $p_2, p_3, \ldots, p_n \overset{\text{i.i.d.}}{\sim} \text{Unif}_{[0,1]}$ and set $P_i = 1/i^{2p_i + 0.25}$.

For each experiment, we run eight setups, each with different combinations of sampler policies, initialization policies of the final phase, and the value of regularization coefficient $\lambda$. For the warm-up phases we set $n = 10$ and for final phases $n = 100$.

**Results.** Figure 4 (with its full view Figure 5), Figure 6, Figure 7, along with Figure 1 (with seed 2018011309) show four experiments of BCP. They shared a learning rate of 0.2, batch size of 100 per step in horizon, final $n = 100$ and warm-up $n = 10$ (if applied curriculum learning). [2]

The experiment in Figure 4 was done in the classical BCP environment, i.e., all permutations have probability $1/n!$ to be sampled. Experiments Figure 1, Figure 6 and Figure 7 were done with other distributions: the only differences are the random seeds, which we fixed and used to generate $P_i$s for reproducibility.

The experiment of classical BCP was run until the direct training of $n = 100$ converges, while all other experiments were run to a maximum episode of 30000 (hence sample number of $THb = 30000 \times 100 \times 100 = 3 \times 10^8$).

The optimal policy was derived from dynamic programming.

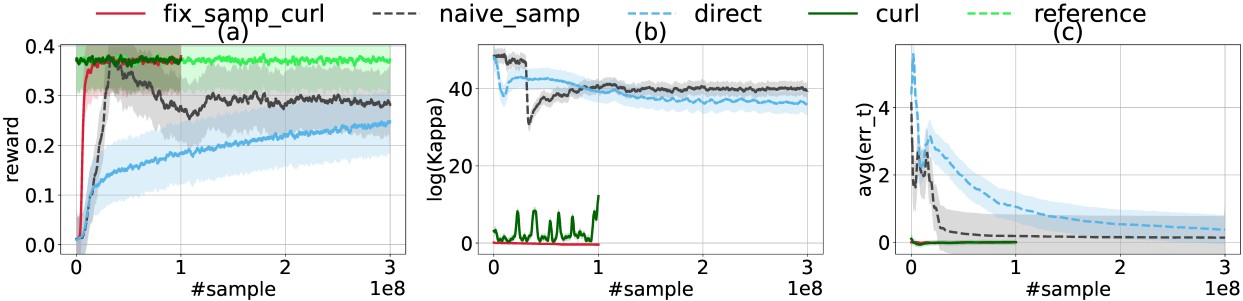

Figure 4: Classical BCP, truncated to $3 \times 10^8$ samples.

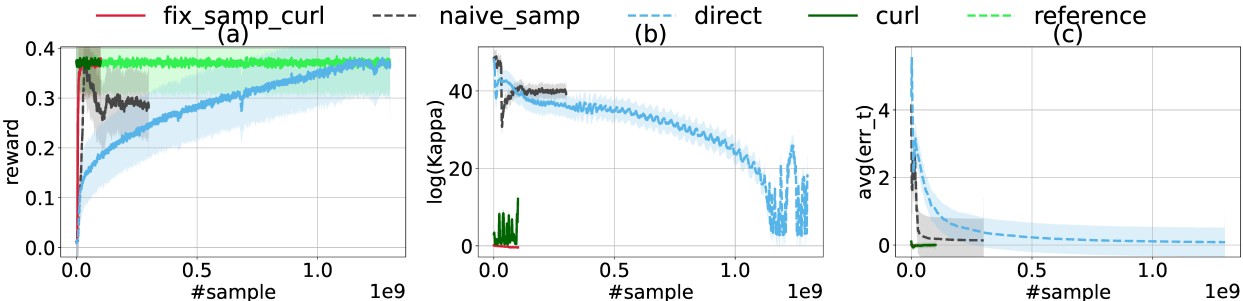

Figure 5: Classical BCP, full view.

---

[2]All the four trainings shown in the figures have their counterparts with regularization ($\lambda = 0.01$). Check the supplementary files and use TensorBoard for visualization.

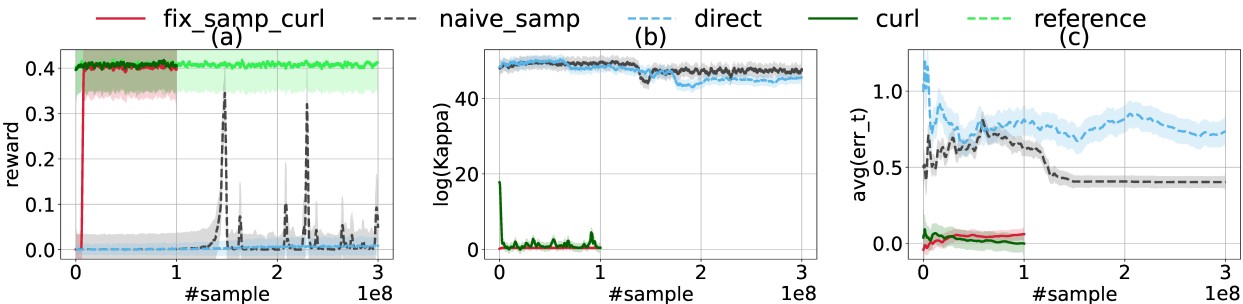

Figure 6: BCP, with seed 20000308.

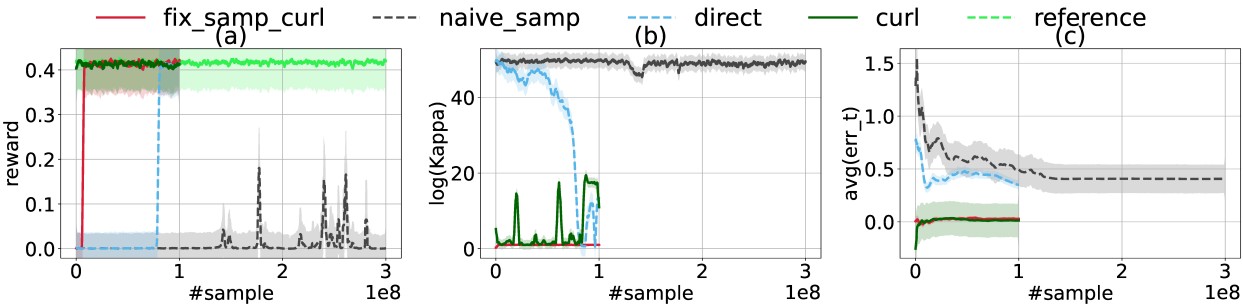

Figure 7: BCP, with seed 19283746.

## C.2 Online Knapsack (decision version, OKD)

**State and action spaces.** The states are represented as

$$s = \left( \frac{i}{n}, s_i, v_i, \frac{\sum_{j=1}^{i-1} x_j s_j}{B}, \frac{\sum_{j=1}^{i-1} x_j v_j}{V} \right),$$

where $x_j = \mathbb{1}[\text{item } j \text{ was successfully chosen}]$ for $1 \leqslant j \leqslant i - 1$ (in the instance). There is an additional terminal state $g = (0, 0, 0, 0, 0)$. For each state (including $g$ for simplicity), the agent can either accept or reject.

**Transition and reward.** The transition is implied by the definition of the problem. Any action in terminal state $g$ leads back to $g$. The item is successfully chosen if and only if the agent accepts and the budget is sufficient. A reward of 1 is given only the first time $\sum_{j=1}^{i} x_i v_i \geqslant V$, and then the state goes to $g$. For all other cases, reward is 0.

**Feature mapping.** Suppose the state is $(f, s, v, r, q)$. We set a degree $d_0$ and the feature mapping is constructed as the collection of polynomial bases with degree less than $d_0$ ($d = d_0^5$): $\phi(f, s, v, r, q) = (f^{i_f} s^{i_s} v^{i_v} r^{i_r} q^{i_q})_{i_f, i_s, i_v, i_r, i_q}$ where $i_\clubsuit \in \{0, 1, \ldots, d_0 - 1\}$.

**LMDP distribution.** In Section 3.2 the values and sizes are sampled from $F_v$ and $F_s$. If $F_v$ or $F_s$ is not $\text{Unif}_{[0,1]}$, we model the distribution as: first set a granularity $gran$ and take $gran$ numbers $p_1, p_2, \ldots, p_{gran} \overset{\text{i.i.d.}}{\sim} \text{Unif}_{[0,1]}$. $p_i$ represents the (unnormalized) probability that $x \in ((i-1)/gran, i/gran)$. To sample, we take $i \sim \text{Multinomial}(p_1, p_2, \ldots, p_{gran})$ and return $x \sim (i - 1 + \text{Unif}_{[0,1]})/gran$.

For each experiment, we ran four setups, each with different combinations of sampler policies and initialization policies of the final phase. For the warm-up phases $n = 10$ and for final phases we set $n = 100$ in all

experiments, while $B$ and $V$ vary. In one experiment it satisfies that $B/n$ are close for warm-up and final, and $V/B$ increases from warm-up to final.

**Results.** Figure 8, Figure 9, along with Figure 2 (with $F_v = F_s = \text{Unif}_{[0,1]}$) show three experiments of OKD. They shared a learning rate of 0.1, batch size of 100 per step in horizon, final $n = 100$ and warm-up $n = 10$ (if applied curriculum learning).

Experiments in Figure 8 and Figure 9 were done with other value and size distributions: the only differences are the random seeds, which we fixed and used to generate $F_v$ and $F_s$ for reproducibility.

All experiments were run to a maximum episode of 50000 (hence sample number of $THb = 50000 \times 100 \times 100 = 5 \times 10^8$).

The reference policy is a bang-per-buck algorithm (Section 3.1 of Kong et al. (2019)): given a threshold $r$, accept $i$-th item if $v_i/s_i \geqslant r$. We searched for the optimal $r$ with respect to Online Knapsack because we found that in general the reward is unimodal to $r$ and contains no "plain area", so we can easily apply ternary search (the reward of OKD contains "plain area").

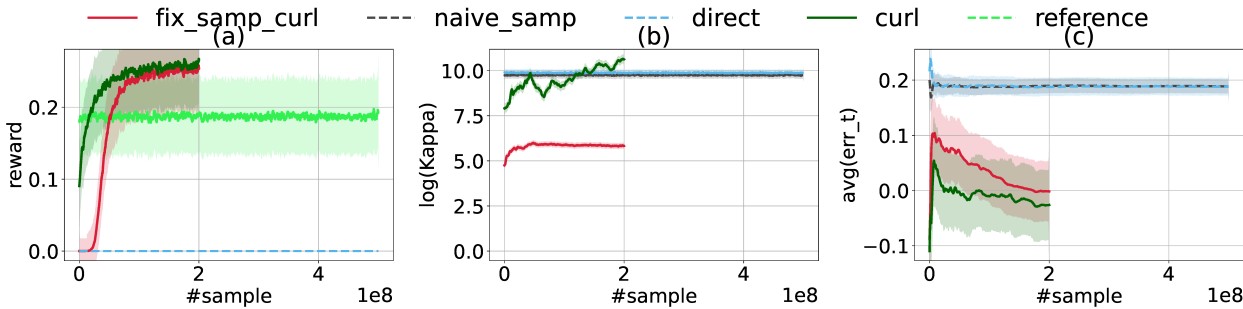

Figure 8: OKD, with seed 2018011309.

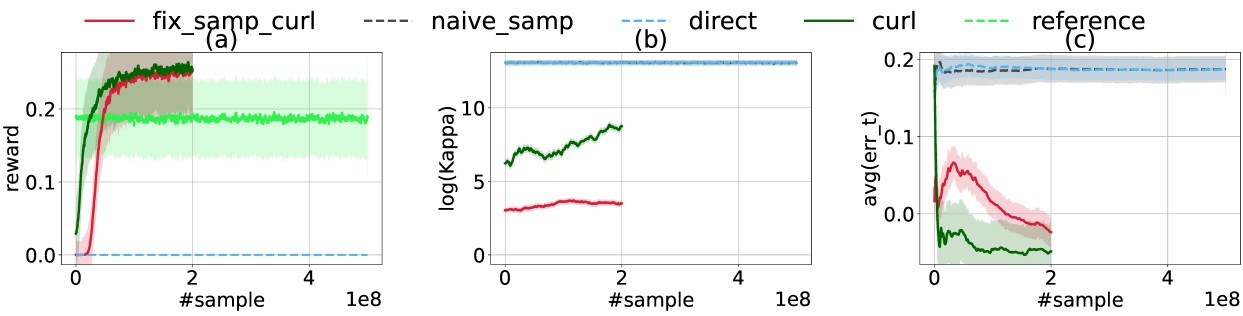

Figure 9: OKD, with seed 20000308.

## C.3 AdWords (decision version, ADW)

**State and action spaces.** The states are represented as

$$s = \left( \frac{j}{m}, v_{1,j}, v_{2,j}, \ldots, v_{n,j}, B_1, B_2, \ldots, B_n, \frac{V_j}{V} \right),$$

where $V_j$ is equal to the total revenue up until now. There is an additional terminal state $g = 0^{2n+2}$. For each state (including $g$ for simplicity), the agent has $n+1$ actions, with 0 representing not assigning the slot and $1, \ldots, n$ representing assigning to the corresponding advertiser.

**Transition and reward.** The transition is implied by the definition of the problem. Any action in terminal state $g$ leads back to $g$. The slot $j$ is successfully assigned to advertiser $i$ if and only if the action is $i$ and $B_i \geqslant v_{i,j}$. The next state is then with $B_i \leftarrow B_i - v_{i,j}$ and $V_j \leftarrow V_j + v_{i,j}$. A reward of 1 is given only the first time $V_j + v_{i,j} \geqslant V$, and then the state goes to $g$. For all other cases, reward is 0.

**Feature mapping.** The feature design in ADW is a bit tricky, since the state dimension is super large. We simplify the setting by assuming all the advertisers are symmetric, so we design a function $\phi$ and for action $1 \leqslant i \leqslant n$,

$$\phi_{s,i} = \phi\left(\frac{j}{m}, v_{i,j}, B_i, \frac{V_j}{V}\right),$$

and

$$\phi_{s,0} = \phi\left(\frac{j}{m}, 0, 0, \frac{V_j}{V}\right).$$

Actually, not assigning the slot is equal to assigning the slot to a virtual advertiser with value 0.

We set a degree $d_0$ and $\phi(f, v, B, q)$ is constructed as the collection of polynomial bases with degree less than $d_0$ $(d = d_0^4)$: $\phi(f, v, B, q) = (f^{i_f} v^{i_v} B^{i_B} q^{i_q})_{i_f, i_v, i_B, i_q}$ where $i_\clubsuit \in \{0, 1, \ldots, d_0 - 1\}$.

**LMDP distribution.** In Section 3.3 the values $v_{i,j}$ are sampled from $F_i$. If $F_i$ is not $\text{Unif}_{[0,1]}$, we model the distribution in the same manner as in OKD. For each experiment, we ran four setups, each with different combinations of sampler policies and initialization policies of the final phase.

In the experiment depicted in Figure 3: For the warm-up phases we set $(n, m) = (3, 6)$ and $V = 2.7$. For final phases we set $(n, m) = (10, 20)$ and $V = 9$. The distributions are parameterized random ones with $gran = 10$.

In the experiment depicted in Figure 10: For the warm-up phases we set $(n, m) = (3, 6)$ and $V = 2.64$. For final phases we set $(n, m) = (8, 32)$ and $V = 7.04$. The distributions are specially designed distributions, with probability $p$ it has a 0.4 value, and the rest $1 - p$ mass is random on $(0.6, 1)$. This distribution type has a special near optimal policy class: either pick two 0.4, or pick anything in $(0.8, 1)$.

**Results.** Figure 3 and Figure 10 are experiments of ADW. They shared a learning rate of 0.1 and batch size of 100 per step in horizon.

The reference policy is obtained by first running a curriculum learning, then using the learned policy as the reference policy. This is because after we simplify the feature representation, we need to compare with the near optimal policy inside this restricted policy / feature class.

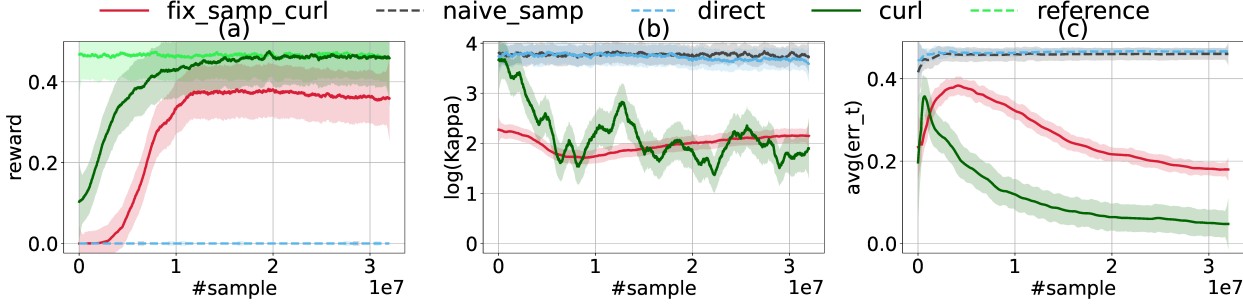

Figure 10: ADW, with seed 19260817 and special distributions.

# D    Technical Details and Lemmas

## D.1    Natural Policy Gradient for LMDP

This section is a complement to Section 5. We give details about the correctness of Natural Policy Gradient for LMDP.

Theorem 12 is the finite-horizon Policy Gradient Theorem for LMDP, which takes the mixing weight $\{w_m\}$ into consideration.

According to Agarwal et al. (2021), the unconstrained, full-information NPG update weight satisfies $F(\theta_t)g_t = \nabla_\theta V^{t,\lambda}$. Lemma 13 and Lemma 14 together show that: it is equivalent to finding a minimizer of the fitting compatible function approximation loss (Definition 8).

**Theorem 12** (Policy Gradient Theorem for LMDP). *For any policy $\pi_\theta$ parameterized by $\theta$, and any $1 \leqslant m \leqslant M$,*

$$\nabla_\theta \left( \mathbb{E}_{s_0 \sim \nu_m} \left[ V_{m,H}^{\pi_\theta,\lambda}(s_0) \right] \right) = \sum_{h=1}^{H} \mathbb{E}_{s,a \sim d_{m,H-h}^{\theta}} \left[ Q_{m,h}^{\pi_\theta,\lambda}(s,a) \nabla_\theta \ln \pi_\theta(a|s) \right].$$

*As a result,*

$$\nabla_\theta V^{\pi_\theta,\lambda} = \sum_{m=1}^{M} w_m \sum_{h=1}^{H} \mathbb{E}_{s,a \sim d_{m,H-h}^{\theta}} \left[ Q_{m,h}^{\pi_\theta,\lambda}(s,a) \nabla_\theta \ln \pi_\theta(a|s) \right].$$

*Proof.* For any $1 \leqslant h \leqslant H$ and $s \in \mathcal{S}$, since $V_{m,h}^{\pi_\theta,\lambda}(s) = \sum_{a \in \mathcal{A}} \pi_\theta(a|s) Q_{m,h}^{\pi_\theta,\lambda}(s,a)$, we have

$$\nabla_\theta V_{m,h}^{\pi_\theta,\lambda}(s) = \sum_{a \in \mathcal{A}} \left( Q_{m,h}^{\pi_\theta,\lambda}(s,a) \nabla_\theta \pi_\theta(a|s) + \pi_\theta(a|s) \nabla_\theta Q_{m,h}^{\pi_\theta,\lambda}(s,a) \right).$$

Hence

$$\sum_{h=1}^{H} \sum_{s \in \mathcal{S}} d_{m,H-h}^{\theta}(s) \nabla_\theta V_{m,h}^{\pi_\theta,\lambda}(s) = \sum_{h=1}^{H} \sum_{s \in \mathcal{S}} d_{m,H-h}^{\theta}(s) \sum_{a \in \mathcal{A}} \left( Q_{m,h}^{\pi_\theta,\lambda}(s,a) \nabla_\theta \pi_\theta(a|s) + \pi_\theta(a|s) \nabla_\theta Q_{m,h}^{\pi_\theta,\lambda}(s,a) \right)$$

$$= \sum_{h=1}^{H} \sum_{s \in \mathcal{S}} d_{m,H-h}^{\theta}(s) \sum_{a \in \mathcal{A}} \pi_\theta(a|s) Q_{m,h}^{\pi_\theta,\lambda}(s,a) \nabla_\theta \ln \pi_\theta(a|s)$$

$$+ \sum_{h=1}^{H} \sum_{s \in \mathcal{S}} d_{m,H-h}^{\theta}(s) \sum_{a \in \mathcal{A}} \pi_\theta(a|s) \nabla_\theta Q_{m,h}^{\pi_\theta,\lambda}(s,a)$$

$$= \sum_{h=1}^{H} \mathbb{E}_{s,a \sim d_{m,H-h}^{\theta}} \left[ Q_{m,h}^{\pi_\theta,\lambda}(s,a) \nabla_\theta \ln \pi_\theta(a|s) \right]$$

$$+ \sum_{h=1}^{H} \sum_{s \in \mathcal{S}} d_{m,H-h}^{\theta}(s) \sum_{a \in \mathcal{A}} \pi_\theta(a|s) \nabla_\theta Q_{m,h}^{\pi_\theta,\lambda}(s,a).$$

Next we focus on the second term. From the Bellman equation,

$$\nabla_\theta Q_{m,h}^{\pi_\theta,\lambda}(s,a) = \nabla_\theta \left( r_\theta(s,a) - \lambda \ln \pi_\theta(a|s) + \sum_{s' \in \mathcal{S}} P(s'|s,a) V_{m,h-1}^{\pi_\theta,\lambda}(s') \right)$$

$$= -\lambda \nabla_\theta \ln \pi_\theta(a|s) + \sum_{s' \in \mathcal{S}} P(s'|s,a) \nabla_\theta V_{m,h-1}^{\pi_\theta,\lambda}(s').$$

Particularly, $\nabla_\theta Q_{i,1}^{\pi,\lambda}(s,a) = -\lambda \nabla_\theta \ln \pi_\theta(a|s)$. So

$$\sum_{h=1}^{H} \sum_{s \in \mathcal{S}} d_{m,H-h}^{\theta}(s) \sum_{a \in \mathcal{A}} \pi_\theta(a|s) \nabla_\theta Q_{m,h}^{\pi_\theta,\lambda}(s,a)$$

$$
= \sum_{h=1}^{H} \sum_{s \in \mathcal{S}} d_{m,H-h}^{\theta}(s) \sum_{a \in \mathcal{A}} \pi_\theta(a|s) \left( -\lambda \nabla_\theta \ln \pi_\theta(a|s) + \sum_{s' \in \mathcal{S}} P(s'|s,a) \nabla_\theta V_{m,h-1}^{\pi_\theta,\lambda}(s') \right)
$$

$$
= -\lambda \sum_{h=1}^{H} \sum_{s \in \mathcal{S}} d_{m,H-h}^{\theta}(s) \underbrace{\sum_{a \in \mathcal{A}} \nabla_\theta \pi_\theta(a|s)}_{=\mathbf{0}} + \sum_{h=2}^{H} \sum_{s' \in \mathcal{S}} \nabla_\theta V_{m,h-1}^{\pi_\theta,\lambda}(s') \underbrace{\sum_{s \in \mathcal{S}} d_{m,H-h}^{\theta}(s) \sum_{a \in \mathcal{A}} \pi_\theta(a|s) P(s'|s,a)}_{=d_{m,H-h+1}^{\theta}(s')}
$$

$$
= \sum_{h=2}^{H} \sum_{s' \in \mathcal{S}} d_{m,H-h+1}^{\theta}(s') \nabla_\theta V_{m,h-1}^{\pi_\theta,\lambda}(s')
$$

$$
= \sum_{h=1}^{H} \sum_{s \in \mathcal{S}} d_{m,H-h}^{\theta}(s) \nabla_\theta V_{m,h}^{\pi_\theta,\lambda}(s) - \sum_{s_0 \in \mathcal{S}} \nu_m(s_0) \nabla_\theta V_{m,H}^{\pi_\theta,\lambda}(s_0),
$$

where we used the definition of $d$ and $\nu_m$. So by rearranging the terms, we complete the proof. $\qquad\square$

**Lemma 13.** *Suppose* $\Gamma \in \mathbb{R}^{n \times m}, D = \mathrm{diag}(d_1, d_2, \ldots, d_m) \in \mathbb{R}^{m \times m}$ *where* $d_i \geqslant 0$ *and* $q \in \mathbb{R}^m$, *then* $x = (\Gamma D \Gamma^\top)^\dagger \Gamma D q$ *is a solution to the equation* $\Gamma D \Gamma^\top x = \Gamma D q$.

*Proof.* Denote $D^{1/2} = \mathrm{diag}(\sqrt{d_1}, \sqrt{d_2}, \ldots, \sqrt{d_m}), P = \Gamma D^{1/2}, p = D^{1/2}q$, then the equation is reduced to $PP^\top x = Pp$. Suppose the singular value decomposition of $P$ is $U \Sigma V^\top$ where $U \in \mathbb{R}^{n \times n}, \Sigma \in \mathbb{R}^{n \times m}, V \in \mathbb{R}^{m \times m}$ where $U$ and $V$ are unitary, and singular values are $\sigma_1, \sigma_2, \ldots, \sigma_k$. So $PP^\top = U(\Sigma \Sigma^\top) U^\top$ and $(PP^\top)^\dagger = U(\Sigma \Sigma^\top)^\dagger U^\top$. Notice that

$$
\Sigma \Sigma^\top = \mathrm{diag}(\sigma_1^2, \sigma_2^2, \ldots, \sigma_k^2, 0, \ldots, 0) \in \mathbb{R}^{n \times n},
$$

we can then derive the pseudo-inverse of this particular diagonal matrix as

$$
(\Sigma \Sigma^\top)^\dagger = \mathrm{diag}(\sigma_1^{-2}, \sigma_2^{-2}, \ldots, \sigma_k^{-2}, 0, \ldots, 0).
$$

It is then easy to verify that $(\Sigma \Sigma^\top)(\Sigma \Sigma^\top)^\dagger \Sigma = \Sigma$. Finally,

$$
\begin{aligned}
PP^\top x &= (PP^\top)[(PP^\top)^\dagger Pp] \\
&= U(\Sigma \Sigma^\top) U^\top U(\Sigma \Sigma^\top)^\dagger U^\top U \Sigma V^\top p \\
&= U(\Sigma \Sigma^\top)(\Sigma \Sigma^\top)^\dagger \Sigma V^\top p \\
&= U \Sigma V^\top p \\
&= Pp.
\end{aligned}
$$

This completes the proof. $\qquad\square$

**Lemma 14** (NPG Update Rule)**.** *The update rule* $\theta \leftarrow \theta + \eta F(\theta)^\dagger \nabla_\theta V^{\pi_\theta,\lambda}$ *where*

$$
F(\theta) = \sum_{m=1}^{M} w_m \sum_{h=1}^{H} \mathbb{E}_{s,a \sim d_{m,H-h}^{\theta}} \left[ (\nabla_\theta \ln \pi_\theta(a|s))^{\otimes} \right]
$$

*is equivalent to* $\theta \leftarrow \theta + \eta g^\star$, *where* $g^\star$ *is a minimizer of the function*

$$
L(g) = \sum_{m=1}^{M} w_m \sum_{h=1}^{H} \mathbb{E}_{s,a \sim d_{m,H-h}^{\theta}} \left[ \left( A_{m,h}^{\pi_\theta,\lambda}(s,a) - g^\top \nabla_\theta \ln \pi_\theta(a|s) \right)^2 \right].
$$

*Proof.*

$$
\nabla_g L(g) = -2 \sum_{m=1}^{M} w_m \sum_{h=1}^{H} \mathbb{E}_{s,a \sim d_{m,H-h}^{\theta}} \left[ \left( A_{m,h}^{\pi_\theta,\lambda}(s,a) - g^\top \nabla_\theta \ln \pi_\theta(a|s) \right) \nabla_\theta \ln \pi_\theta(a|s) \right].
$$

Suppose $g^\star$ is any minimizer of $L(g)$, we have $\nabla_g L(g^\star) = \mathbf{0}$, hence

$$\sum_{m=1}^{M} w_m \sum_{h=1}^{H} \mathbb{E}_{s,a \sim d_{m,H-h}^\theta} \left[ \left( g^{\star\top} \nabla_\theta \ln \pi_\theta(a|s) \right) \nabla_\theta \ln \pi_\theta(a|s) \right]$$

$$= \sum_{m=1}^{M} w_m \sum_{h=1}^{H} \mathbb{E}_{s,a \sim d_{m,H-h}^\theta} \left[ A_{m,h}^{\pi_\theta,\lambda}(s,a) \nabla_\theta \ln \pi_\theta(a|s) \right]$$

$$= \sum_{m=1}^{M} w_m \sum_{h=1}^{H} \mathbb{E}_{s,a \sim d_{m,H-h}^\theta} \left[ Q_{m,h}^{\pi_\theta,\lambda}(s,a) \nabla_\theta \ln \pi_\theta(a|s) \right].$$

Since $(u^\top v)v = (vv^\top)u$, then

$$F(\theta)g^\star = \nabla_\theta V^{\pi_\theta,\lambda}.$$

Now we assign $1, 2, \ldots, MHSA$ as indices to all $(m, h, s, a) \in \{1, \ldots, M\} \times \{1, \ldots, H\} \times \mathcal{S} \times \mathcal{A}$, and set

$$\gamma_j = \nabla_\theta \ln \pi_\theta(a|s),$$
$$d_j = w_m d_{m,H-h}^\theta(s,a),$$
$$q_j = Q_{m,h}^{\pi_\theta,\lambda}(s,a),$$

where $j$ is the index assigned to $(m, h, s, a)$. Then $F(\theta) = \Phi D \Phi^\top$ and $\nabla_\theta V^\theta = \Phi D q$ where

$$\Gamma = [\gamma_1, \gamma_2, \ldots, \gamma_{MHSA}] \in \mathbb{R}^{d \times MHSA},$$
$$D = \mathrm{diag}(d_1, d_2, \ldots, d_{MHSA}) \in \mathbb{R}^{MHSA \times MHSA},$$
$$q = [q_1, q_2, \ldots, q_{MHSA}]^\top \in \mathbb{R}^{MHSA}.$$

We now conclude the proof by utilizing Lemma 13. $\qquad\square$

## D.2 Auxiliary lemmas used in the main results

**Lemma 15** (Performance Difference Lemma). *For any two policies $\pi_1$ and $\pi_2$, and any $1 \leqslant m \leqslant M$,*

$$\mathbb{E}_{s_0 \sim \nu_m} \left[ V_{m,H}^{\pi_1,\lambda}(s_0) - V_{m,H}^{\pi_2,\lambda}(s_0) \right] = \sum_{h=1}^{H} \mathbb{E}_{s,a \sim d_{m,H-h}^{\pi_1}} \left[ A_{m,h}^{\pi_2,\lambda}(s,a) + \lambda \ln \frac{\pi_2(a|s)}{\pi_1(a|s)} \right].$$

*As a result,*

$$V^{\pi_1,\lambda} - V^{\pi_2,\lambda} = \sum_{m=1}^{M} w_m \sum_{h=1}^{H} \mathbb{E}_{s,a \sim d_{m,H-h}^{\pi_1}} \left[ A_{m,h}^{\pi_2,\lambda}(s,a) + \lambda \ln \frac{\pi_2(a|s)}{\pi_1(a|s)} \right].$$

*Proof.* First we fix $s_0$. By definition of the value function, we have

$$V_{m,H}^{\pi_1,\lambda}(s_0) - V_{m,H}^{\pi_2,\lambda}(s_0)$$

$$= \mathbb{E}\left[ \sum_{h=0}^{H-1} r_m(s_h, a_h) - \lambda \ln \pi_1(a_h|s_h) \,\middle|\, \mathcal{M}_m, \pi_1, s_0 \right] - V_{m,H}^{\pi_2,\lambda}(s_0)$$

$$= \mathbb{E}\left[ \sum_{h=0}^{H-1} r_m(s_h, a_h) - \lambda \ln \pi_1(a_h|s_h) + V_{m,H+1-h}^{\pi_2,\lambda}(s_{h+1}) - V_{m,H-h}^{\pi_2,\lambda}(s_h) \,\middle|\, \mathcal{M}_m, \pi_1, s_0 \right]$$

$$= \mathbb{E}\left[ \sum_{h=0}^{H-1} \mathbb{E}\left[ r_m(s_h, a_h) - \lambda \ln \pi_2(a_h|s_h) + V_{m,H+1-h}^{\pi_2,\lambda}(s_{h+1}) \,\middle|\, \mathcal{M}_m, \pi_2, s_h, a_h \right] \,\middle|\, \mathcal{M}_m, \pi_1, s_0 \right]$$

$$+ \mathbb{E}\left[\sum_{h=0}^{H-1} -V_{m,H-h}^{\pi_2,\lambda}(s_h) + \lambda \ln \frac{\pi_2(a_h|s_h)}{\pi_1(a_h|s_h)} \;\middle|\; \mathcal{M}_m, \pi_1, s_0\right],$$

where the last step uses law of iterated expectations. Since

$$\mathbb{E}\left[r_m(s_h, a_h) - \lambda \ln \pi_2(a_h|s_h) + V_{m,H+1-h}^{\pi_2,\lambda}(s_{h+1}) \;\middle|\; \mathcal{M}_m, \pi_2, s_h, a_h\right] = Q_{m,H-h}^{\pi_2,\lambda}(s_h, a_h),$$

we have

$$V_{m,H}^{\pi_1,\lambda}(s_0) - V_{m,H}^{\pi_2,\lambda}(s_0) = \mathbb{E}\left[\sum_{h=0}^{H-1} Q_{m,H-h}^{\pi_2,\lambda}(s_h, a_h) - V_{m,H-h}^{\pi_2,\lambda}(s_h) + \lambda \ln \frac{\pi_2(a_h|s_h)}{\pi_1(a_h|s_h)} \;\middle|\; \mathcal{M}_m, \pi_1, s_0\right]$$

$$= \mathbb{E}\left[\sum_{h=0}^{H-1} A_{m,H-h}^{\pi_2,\lambda}(s_h, a_h) + \lambda \ln \frac{\pi_2(a_h|s_h)}{\pi_1(a_h|s_h)} \;\middle|\; \mathcal{M}_m, \pi_1, s_0\right].$$

By taking expectation over $s_0$, we have

$$\mathbb{E}_{s_0 \sim \nu_m}\left[V_{m,H}^{\pi_1,\lambda}(s_0) - V_{m,H}^{\pi_2,\lambda}(s_0)\right] = \mathbb{E}\left[\sum_{h=0}^{H-1} A_{m,H-h}^{\pi_2,\lambda}(s_h, a_h) + \lambda \ln \frac{\pi_2(a_h|s_h)}{\pi_1(a_h|s_h)} \;\middle|\; \mathcal{M}_m, \pi_1\right]$$

$$= \sum_{h=0}^{H-1} \sum_{(s,a)\in\mathcal{S}\times\mathcal{A}} d_{m,h}^{\pi_1}(s,a) \left(A_{m,H-h}^{\pi_2,\lambda}(s,a) + \lambda \ln \frac{\pi_2(a|s)}{\pi_1(a|s)}\right).$$

The proof is completed by reversing the order of $h$. $\qquad\square$

**Lemma 16** (Lyapunov Drift). *Recall definitions in Definitions 5 and 9. We have that:*

$$\Phi(\pi_{t+1}) - \Phi(\pi_t) \leq -\eta\lambda\Phi(\pi_t) + \eta\mathrm{err}_t - \eta\left(V^{\star,\lambda} - V^{t,\lambda}\right) + \frac{\eta^2 B^2 \|g_t\|_2^2}{2}.$$

*Proof.* Denote $\Phi_t := \Phi(\pi_t)$. This proof follows a similar manner as in that of Lemma 6 in Cayci et al. (2021). By smoothness (see Remark 6.7 in Agarwal et al. (2021)),

$$\ln \frac{\pi_t(a|s)}{\pi_{t+1}(a|s)} \leq (\theta_t - \theta_{t+1})^\top \nabla_\theta \ln \pi_t(a|s) + \frac{B^2}{2}\|\theta_{t+1} - \theta_t\|_2^2$$

$$= -\eta g_t^\top \nabla_\theta \ln \pi_t(a|s) + \frac{\eta^2 B^2 \|g_t\|_2^2}{2}.$$

By the definition of $\Phi$,

$$\Phi_{t+1} - \Phi_t = \sum_{m=1}^{M} w_m \sum_{h=1}^{H} \mathbb{E}_{(s,a)\sim d_{m,H-h}^\star}\left[\ln \frac{\pi_t(a|s)}{\pi_{t+1}(a|s)}\right]$$

$$\leq -\eta \sum_{m=1}^{M} w_m \sum_{h=1}^{H} \mathbb{E}_{(s,a)\sim d_{m,H-h}^\star}\left[g_t^\top \nabla_\theta \ln \pi_t(a|s)\right] + \frac{\eta^2 B^2 \|g_t\|_2^2}{2}.$$

By the definition of $\mathrm{err}_t$, Lemma 15 and again the definition of $\Phi$, we finally have

$$\Phi_{t+1} - \Phi_t \leq \eta \sum_{m=1}^{M} w_m \sum_{h=1}^{H} \mathbb{E}_{(s,a)\sim d_{m,H-h}^\star}\left[A_{m,h}^{t,\lambda}(s,a) - g_t^\top \nabla_\theta \ln \pi_t(a|s)\right]$$

$$- \eta \sum_{m=1}^{M} w_m \sum_{h=1}^{H} \mathbb{E}_{(s,a)\sim d_{m,H-h}^\star}\left[A_{m,h}^{t,\lambda}(s,a) + \lambda \ln \frac{\pi_t(a|s)}{\pi^\star(a|s)}\right]$$

$$- \eta\lambda \sum_{m=1}^{M} w_m \sum_{h=1}^{H} \mathbb{E}_{(s,a)\sim d_{m,H-h}^\star}\left[\ln \frac{\pi^\star(a|s)}{\pi_t(a|s)}\right] + \frac{\eta^2 B^2 \|g_t\|_2^2}{2}$$

$$= \eta \mathrm{err}_t - \eta \left( V^{\star,\lambda} - V^{t,\lambda} \right) - \eta \lambda \Phi_t + \frac{\eta^2 B^2 \|g_t\|_2^2}{2},$$

which completes the proof. $\qquad\square$

**Lemma 17.** *Recall that $g_t^\star$ is the true minimizer of $L(g; \theta_t, d^t)$ in domain $\mathcal{G}$. $\mathrm{err}_t$ defined in Definition 5 satisfies*

$$\mathrm{err}_t \leqslant \sqrt{H L(g_t^\star; \theta_t, d^\star)} + \sqrt{H \kappa (L(g_t; \theta_t, d^t) - L(g_t^\star; \theta_t, d^t))}.$$

*Proof.* The proof is similar to that of Theorem 6.1 in Agarwal et al. (2021). We make the following decomposition of $\mathrm{err}_t$:

$$\mathrm{err}_t = \underbrace{\sum_{m=1}^{M} w_m \sum_{h=0}^{H-1} \mathbb{E}_{(s,a) \sim d_{m,h}^\star} \left[ A_{m,h}^{t,\lambda}(s,a) - g_t^{\star\top} \nabla_\theta \ln \pi_t(a|s) \right]}_{\text{①}}$$

$$+ \underbrace{\sum_{m=1}^{M} w_m \sum_{h=0}^{H-1} \mathbb{E}_{(s,a) \sim d_{m,h}^\star} \left[ (g_t^\star - g_t)^\top \nabla_\theta \ln \pi_t(a|s) \right]}_{\text{②}}.$$

Since $\sum_{m=1}^{M} w_m \sum_{h=0}^{H-1} \sum_{(s,a) \in \mathcal{S} \times \mathcal{A}} d_{m,h}^\star(s,a) = H$, normalize the coefficients and apply Jensen's inequality, then

$$\text{①} \leqslant \sqrt{\sum_{m=1}^{M} w_m \sum_{h=0}^{H-1} \sum_{(s,a) \in \mathcal{S} \times \mathcal{A}} d_{m,h}^\star(s,a)} \cdot \sqrt{\sum_{m=1}^{M} w_m \sum_{h=0}^{H-1} \mathbb{E}_{(s,a) \sim d_{m,h}^\star} \left[ \left( A_{m,h}^{t,\lambda}(s,a) - g_t^{\star\top} \nabla_\theta \ln \pi_t(a|s) \right)^2 \right]}$$

$$= \sqrt{H L(g_t^\star; \theta_t, d^\star)}.$$

Similarly,

$$\text{②} \leqslant \sqrt{H \sum_{m=1}^{M} w_m \sum_{h=0}^{H-1} \mathbb{E}_{(s,a) \sim d_{m,h}^\star} \left[ ((g_t^\star - g_t)^\top \nabla_\theta \ln \pi_t(a|s))^2 \right]}$$

$$= \sqrt{H \sum_{m=1}^{M} w_m \sum_{h=0}^{H-1} \mathbb{E}_{(s,a) \sim d_{m,h}^\star} \left[ (g_t^\star - g_t)^\top \nabla_\theta \ln \pi_t(a|s) (\nabla_\theta \ln \pi_t(a|s))^\top (g_t^\star - g_t) \right]}$$

$$\overset{\text{(i)}}{=} \sqrt{H \| g_t^\star - g_t \|_{\Sigma_{d^\star}^t}^2}$$

$$\leqslant \sqrt{H \kappa \| g_t^\star - g_t \|_{\Sigma_t}^2},$$

where in (i), for vector $v$, denote $\|v\|_A = \sqrt{v^\top A v}$ for a symmetric positive semi-definite matrix $A$. Due to that $g_t^\star$ minimizes $L(g; \theta_t, d^t)$ over the set $\mathcal{G}$, the first-order optimality condition implies that

$$(g - g_t^\star)^\top \nabla_g L(g_t^\star; \theta_t, d^t) \geqslant 0$$

for any $g$. Therefore,

$$L(g; \theta_t, d^t) - L(g_t^\star; \theta_t, d^t)$$

$$= \sum_{m=1}^{M} w_m \sum_{h=1}^{H} \mathbb{E}_{s,a \sim d_{m,H-h}^t} \left[ \left( A_{m,h}^{t,\lambda}(s,a) - g_t^{\star\top} \nabla \ln \pi_t(a|s) + (g_t^\star - g)^\top \nabla \ln \pi_t(a|s) \right)^2 \right] - L(g_t^\star; \theta_t, d^t)$$

$$= \sum_{m=1}^{M} w_m \sum_{h=1}^{H} \mathbb{E}_{s,a \sim d_{m,H-h}^t} \left[ ((g_t^\star - g)^\top \nabla_\theta \ln \pi_t(a|s))^2 \right]$$

$$+ (g - g_t^\star)^\top \left( -2 \sum_{m=1}^M w_m \sum_{h=1}^H \mathbb{E}_{s,a \sim d_{m,H-h}^t} \left[ \left( A_{m,h}^{t,\lambda}(s,a) - g_t^{\star\top} \nabla_\theta \ln \pi_t(a|s) \right) \nabla_\theta \ln \pi_t(a|s) \right] \right)$$

$$= \|g_t^\star - g\|_{\Sigma_t}^2 + (g - g_t^\star)^\top \nabla_g L(g_t^\star; \theta_t, d^t)$$

$$\geq \|g_t^\star - g\|_{\Sigma_t}^2.$$

So finally we have

$$\text{err}_t \leq \sqrt{H L(g_t^\star; \theta_t, d^\star)} + \sqrt{H \kappa (L(g_t; \theta_t, d^t) - L(g_t^\star; \theta_t, d^t))}.$$

This completes the proof. $\qquad\square$

### D.3 Bounding $\epsilon_{\text{stat}}$

**Lemma 18** (Hoeffding's Inequality). *Suppose $X_1, X_2, \dots, X_n$ are i.i.d. random variables taking values in $[a,b]$, with expectation $\mu$. Let $\bar{X}$ denote their average, then for any $\epsilon \geq 0$,*

$$\mathbb{P}\left( |\bar{X} - \mu| \geq \epsilon \right) \leq 2 \exp\left( -\frac{2n\epsilon^2}{(b-a)^2} \right).$$

**Lemma 19.** *For any policy $\pi$, any state $s \in \mathcal{S}$ and any $U \geq \ln |\mathcal{A}| - 1$,*

$$0 \leq \sum_{a \in \mathcal{A}} \pi(a|s) \ln \frac{1}{\pi(a|s)} - \sum_{a \in \mathcal{A}} \pi(a|s) \min\left\{ \ln \frac{1}{\pi(a|s)}, U \right\} \leq \frac{|\mathcal{A}|}{e^{U+1}}.$$

*Proof.* The first inequality is straightforward, so we focus on the second part. Set $\mathcal{A}' = \{a \in \mathcal{A} : \ln \frac{1}{\pi(a|s)} > U\} = \{a \in \mathcal{A} : \pi(a|s) < \frac{1}{e^U}\}$ and $p = \sum_{a \in \mathcal{A}'} \pi(a|s)$, then

$$\sum_{a \in \mathcal{A}} \pi(a|s) \ln \frac{1}{\pi(a|s)} - \sum_{a \in \mathcal{A}} \pi(a|s) \min\left\{ \ln \frac{1}{\pi(a|s)}, U \right\} = \sum_{a \in \mathcal{A}'} \pi(a|s) \ln \frac{1}{\pi(a|s)} - \sum_{a \in \mathcal{A}'} \pi(a|s) U$$

$$= p \sum_{a \in \mathcal{A}'} \frac{\pi(a|s)}{p} \ln \frac{1}{\pi(a|s)} - pU$$

$$\leq p \ln \left( \sum_{a \in \mathcal{A}'} \frac{\pi(a|s)}{p} \frac{1}{\pi(a|s)} \right) - pU$$

$$\leq p \ln \frac{|\mathcal{A}|}{p} - pU,$$

where the penultimate step comes from concavity of $\ln x$ and Jensen's inequality. Let $f(p) = p \ln \frac{|\mathcal{A}|}{p} - pU$, then $f'(p) = \ln |\mathcal{A}| - U - 1 - \ln p$. Recall that $U \geq \ln |\mathcal{A}| - 1$, so $f(p)$ increases when $p \in (0, \frac{|\mathcal{A}|}{e^{U+1}})$ and decreases when $p \in (\frac{|\mathcal{A}|}{e^{U+1}}, 1)$. Since $f(\frac{|\mathcal{A}|}{e^{U+1}}) = \frac{|\mathcal{A}|}{e^{U+1}}$ we complete the proof. $\qquad\square$

**Lemma 20** (Loss Function Concentration). *If set $\pi_s = None$ and $U \geq \ln |\mathcal{A}| - 1$, then with probability $1 - 2(T+1) \exp\left( -\frac{2N\epsilon^2}{C^2} \right)$, the update weight sequence of Algorithm 3 satisfies: for any $0 \leq t \leq T$,*

$$L(\hat{g}_t; \theta_t, d^{\theta_t}) - L(g_t^\star; \theta_t, d^{\theta_t}) \leq 2\epsilon + \frac{8\lambda G B |\mathcal{A}|}{e^{U+1}},$$

*where*

$$C = 16 H G B [1 + \lambda U + H(1 + \lambda \ln |\mathcal{A}|)] + 4 H G^2 B^2.$$

*If $\pi_s \neq None$ and $\lambda = 0$, then with probability $1 - 2(T+1) \exp\left( -\frac{2N\epsilon^2}{C^2} \right)$, the update weight sequence of Algorithm 3 satisfies: for any $0 \leq t \leq T$,*

$$L(\hat{g}_t; \theta_t, \widetilde{d}^{\pi_s}) - L(g_t^\star; \theta_t, \widetilde{d}^{\pi_s}) \leq 2\epsilon,$$

*where*

$$C = 16H^2GB + 4HG^2B^2.$$

*Proof.* We first prove the $\pi_s =$ None case. For time step $t$, Algorithm 3 samples $HN$ trajectories. Abusing the notation, denote

$$\widehat{F}_t = \frac{1}{N} \sum_{n=1}^{N} \sum_{h=0}^{H-1} \nabla_\theta \ln \pi_\theta(a_{n,h}|s_{n,h}) \left(\nabla_\theta \ln \pi_\theta(a_{n,h}|s_{n,h})\right)^\top,$$

$$\widehat{\nabla}_t = \frac{1}{N} \sum_{n=1}^{N} \sum_{h=0}^{H-1} \widehat{A}_{n,H-h}(s_{n,h}, a_{n,h}) \nabla_\theta \ln \pi_\theta(a_{n,h}|s_{n,h}),$$

$$\widehat{L}(g) = \underbrace{\sum_{m=1}^{M} w_m \sum_{h=1}^{H} \mathbb{E}_{s,a \sim d_{m,H-h}^{\theta_t}} \left[A_{m,h}^{t,\lambda}(s,a)^2\right]}_{①} + \underbrace{g^\top \widehat{F}_t g - 2g^\top \widehat{\nabla}_t}_{②}.$$

Notice that ① is a constant. From Algorithm 3, $\widehat{g}_t$ is the minimizer of ② (hence $\widehat{L}(g)$) inside the ball $\mathcal{G}$. From $\nabla_\theta \ln \pi_\theta(a|s) = \phi(s,a) - \mathbb{E}_{a' \sim \pi_\theta(\cdot|s)}[\phi(s,a')]$, $\|\phi(s,a)\|_2 \leqslant B$, $\|g\|_2 \leqslant G$, we know that $\left|g^\top \nabla_\theta \ln \pi_\theta(a|s)\right| \leqslant 2GB$. So $0 \leqslant g^\top \widehat{F}_t g \leqslant 4HG^2B^2$. From Algorithm 4, we know that any sampled $\widehat{A}$ satisfies $|\widehat{A}| \leqslant 2[1 + \lambda U + H(1 + \lambda \ln|\mathcal{A}|)]$. So $|g^\top \widehat{\nabla}_t| \leqslant 4HGB[1 + \lambda U + H(1 + \lambda \ln|\mathcal{A}|)]$. We first have that

$$-8HGB[1 + \lambda U + H(1 + \lambda \ln|\mathcal{A}|)] \leqslant ② \leqslant 8HGB[1 + \lambda U + H(1 + \lambda \ln|\mathcal{A}|)] + 4HG^2B^2. \qquad (2)$$

To apply any standard concentration inequality, we next need to calculate the expectation of ②. According to Monte Carlo sampling and Lemma 19, for any $1 \leqslant m \leqslant M, 1 \leqslant h \leqslant H$ and $(s,a) \in \mathcal{S} \times \mathcal{A}$, we have

$$A_{m,h}^{t,\lambda}(s,a) - \frac{\lambda|\mathcal{A}|}{e^{U+1}} \leqslant \mathbb{E}\left[\widehat{A}_{m,h}^{t,\lambda}(s,a)\right] \leqslant A_{m,h}^{t,\lambda}(s,a).$$

Denote $\nabla_t$ as the exact policy gradient at time step $t$, then

$$\left|\mathbb{E}\left[g^\top \widehat{\nabla}_t\right] - g^\top \nabla_t\right| \leqslant \|g\|_2 \left\|\mathbb{E}\left[\widehat{\nabla}_t\right] - \nabla_t\right\|_2$$

$$\leqslant \|g\|_2 \cdot H\|\nabla_\theta \ln \pi_\theta(a|s)\|_2 \left\|\mathbb{E}\left[\widehat{A}(s,a)\right] - A(s,a)\right\|_\infty$$

$$\leqslant \frac{2\lambda GB|\mathcal{A}|}{e^{U+1}}.$$

Since Monte Carlo sampling correctly estimates state-action visitation distribution, $\mathbb{E}\left[\widehat{F}_t\right] = F(\theta_t)$. Notice that $g^\top \widehat{F}_t g$ is linear in entries of $\widehat{F}_t$, we have $\mathbb{E}\left[g^\top \widehat{F}_t g\right] = g^\top F(\theta_t)g$. Now we are in the position to show that

$$\left|\mathbb{E}\left[\widehat{L}(g)\right] - L(g)\right| \leqslant \frac{4\lambda GB|\mathcal{A}|}{e^{U+1}}.$$

Hoeffding's inequality (Lemma 18) gives

$$\mathbb{P}\left(\left|\widehat{L}(g) - \mathbb{E}\left[\widehat{L}(g)\right]\right| \geqslant \epsilon\right) \leqslant 2\exp\left(-\frac{2N\epsilon^2}{C^2}\right).$$

where from Equation (2),

$$C = 16HGB[1 + \lambda U + H(1 + \lambda \ln|\mathcal{A}|)] + 4HG^2B^2.$$

After applying union bound for all $t$, with probability $1 - 2(T+1)\exp\left(-\frac{2N\epsilon^2}{C^2}\right)$ the following holds for any $g \in \mathcal{G}$:

$$\left|\widehat{L}(g; \theta_t, d^{\theta_t}) - L(g; \theta_t, d^{\theta_t})\right| \leqslant \epsilon + \frac{4\lambda GB|\mathcal{A}|}{e^{U+1}}.$$

Hence

$$
\begin{aligned}
L(\widehat{g}_t; \theta_t, d^{\theta_t}) &\leqslant \widehat{L}(\widehat{g}_t; \theta_t, d^{\theta_t}) + \epsilon + \frac{4\lambda GB|\mathcal{A}|}{e^{U+1}} \\
&\leqslant \widehat{L}(g_t^\star; \theta_t, d^{\theta_t}) + \epsilon + \frac{4\lambda GB|\mathcal{A}|}{e^{U+1}} \\
&\leqslant L(g_t^\star; \theta_t, d^{\theta_t}) + 2\epsilon + \frac{8\lambda GB|\mathcal{A}|}{e^{U+1}}.
\end{aligned}
$$

For $\pi_s \neq$ None and $\lambda = 0$, we notice that $|\widehat{A}| \leqslant 2H$ and hence $-8H^2GB \leqslant ② \leqslant 8H^2GB + 4HG^2B^2$. Moreover, $\mathbb{E}\left[\widehat{A}_{m,h}^{t,\lambda}(s,a)\right] = A_{m,h}^{t,\lambda}(s,a)$. So by slightly modifying the proof we can get the result. $\qquad\square$

