# OpenReview forum: "Understanding Curriculum Learning in Policy Optimization for Online Combinatorial Optimization"
_TMLR — Accepted by TMLR_

### Review · Reviewer_KbY1 · 2023-08-25

**Summary Of Contributions:**

The paper studies the effectiveness of policy optimization for online combinatorial optimization (CO) problems, which frames online CO as latent Markov Decision Processes (LMDPs). The authors first proved convergence bounds of the natural policy gradient (NPG) for solving LMDPs. Then, the authors shed light on the advantages of curriculum learning: primarily, its potential to establish a robust sampling policy and significantly diminish the distribution shift. In the context of BCP, the authors demonstrates that the distribution shift will have an exponential reduction with curriculum learning, irrespective of whether the curriculum is a randomly generated BCP on a smaller scale. Their insights suggests that Curriculum Learning's primary contribution lies in facilitating a more potent sampling policy, which in turn lowers the relative condition number $\kappa$. To bolster their arguments, the authors also provide empirical results on simple problems.

**Audience:**

Yes

**Broader Impact Concerns:**

Not applicable.

**Claims And Evidence:**

Yes

**Requested Changes:**

See Weaknesses in the above section.

**Strengths And Weaknesses:**

## Strengths:

- The connection of online combinatorial optimization and latent Markov decision processes explored in this work is new and interesting, introducing a fresh perspective to the understandings of CO.

- The theoretical findings underscoring how CL accelerates the reduction of the condition number $\kappa$ serves as a important tool to explain the efficacy and the foundational mechanics of CL. This could be a interesting contribution to the academic community.

- The paper is overll well-written and easy to follow.


## Weakness:

- The set of problems considered in this paper appears to be relatively narrow. There's a doubt regarding the relevance of these problems to the broader machine learning community. An exploration or at least a discussion about the potential to extrapolate these findings to other Reinforcement Learning (RL) problems would enhance its relevance, as CL has been wildely studied and used in many RL algorithms to solve the task, see https://lilianweng.github.io/posts/2020-01-29-curriculum-rl/.

- The paper restricts its theoretical studies majorly to NPG. I wonder if it's possible to replicate similar insights or theoretical results for other RL algorithms, e.g., value based methods.

- It would be beneficial if the authors could include a discussion on how to construct a curriculum for more complex problems. While it may not be very challenging to construct a sequence of curriculum for the problems addressed in the paper, devising a suitable curriculum for more intricate issues remains a significant challenge.


- The theoretical analysis in Section 6 appears somewhat incremental. Given the existing comprehensive understanding of NPG and the similarity between LMDPs and standard MDPs, the analysis seems quite straightforward.

---

> ### Author Response · Authors · 2023-09-20
>
> Thank you for you review!
>
> **Weaknesses:**
>
> - **Relevance to ML community:** In this paper, we present the novel finding that curriculum learning helps reduce the relative condition number $\kappa$. Combined with convergence rate of NPG, we provide a theoretical insight regarding the effectiveness of CL. The relative condition number also presents in offline RL (https://arxiv.org/abs/2107.06226), so our result may have broader impacts.
>
> - **Replicate theoretical results for other RL algorithms:** This is an insightful question. For example in linear function approximation (https://arxiv.org/abs/1907.05388), there is a $\Lambda_h$ which is the same form as $\Sigma_t$ in our paper. Curriculum learning has the potential to yield a better bound of $\sum_j \phi_j^\top (\Lambda_{j-1})^{-1} \phi_j$.
>
> - **Discussion on the curriculum design:** The design is problem-dependent. For the problems described in our paper (best choice problem, online knapsack, online matching) as well as any similar problems (online load balancing, online set cover, etc.), we can use $n$, the sequence length of online decision-making, to represent the difficulty. A high-level point is that we first find the hyperparameters controlling the difficulty of the problem, e.g., the sequence length, the action space size, the number of interaction steps, then reduce these hyperparameters to construct a smaller scale and simpler problem.
>
> - **Theoretical results:** Our result of NPG for LMDP covers most practical situations: sample-based, with entropy regularization and batched updates of weight (line 14 of Algorithm 3). At the time of submission, no previous work provided a unified analysis for these situations to our knowledge. Also, batched updates instead of successive projected gradient ascent (as [Agarwal et al., 2021]) is often used in practical training. Another important aspect of our theoretical findings is that we identify the relative condition number $\kappa$ as a key factor showing effectiveness of curriculum learning. Our theoretical results are not incremental.

---

### Review · Reviewer_d4qr · 2023-09-06

**Summary Of Contributions:**

This paper studies Combinatorial Optimization (CO) problems combined with curriculum learning. The authors study the performance of modeling with problem as an LMDP and using NPG to solve this problem. In order to motivate and study the performance, the authors consider several case studies and analyze the performance of their algorithm applied to these cases.
The authors also provide a theoretical analysis for the convergence of their algorithm.

**Audience:**

Yes

**Broader Impact Concerns:**

The paper is mainly mathematical and a simulation based on synthetic data. I believe broader impact is not necessary.

**Claims And Evidence:**

No

**Requested Changes:**

- Can you add comparison of your result with the best method which does not use RL for solving these CO problems?

**Strengths And Weaknesses:**

- The main weakness of the paper is their theoretical result. In particular, Theorem 6 is essentially the same as the results in Agarwal et al 2021. Also, since Theorem 7 only provides an upper bound, we cannot argue for improved performance of curriculum learning based on Theorem 7.
- Another issue is the experimental results. In particular, the performance of the dashed line is unrealistically bad. We observe flat zero reward for the baseline after 1e7 samples. Did the authors make sure the baseline is tuned carefully enough?

---

> ### Author Response · Authors · 2023-09-20
>
> Thank you for your review!
>
> **Weaknesses:**
>
> - **Theoretical results:** Our result of "NPG for LMDP" (Algorithm 3 and Theorem 6 in the appendix) covers most practical situations: sample-based, with entropy regularization and batched updates of weight (line 14 of Algorithm 3). At the time of submission, no previous work provided a unified analysis for these situations to our knowledge. Also, batched updates instead of successive projected gradient ascent (as [Agarwal et al., 2021]) is often used in practical training. Theorem 7 states that we can construct examples to ensure curriculum learning has benefits. The $\kappa$ in the theorem are tight up to a constant factor of $2$.
>
> - **Experimental results:** uning the baseline was not necessary, because NPG algorithms are guaranteed to converge in a finite episode, and we also did not tune the curriculum learning policies. The behavior of the dashed lines could be described in the following way: first it stays at zero for a significant number of samples, then it suddenly spikes and converges quickly. We plot most figures within the range of $10^8$ samples, however, this sample size is insufficient to observe spikes in the dashed lines. Thus, readers may observe zero performance from the baselines. Please refer to Figure 5 for a full view of BCP. Figure 7 also shows a typical behavior of the blue dashed line.
>
> **Requested changes:**
>
> - **Comparison with non-RL algorithms:** They are included in the figures as "reference policies". All these policies were described in Kong et al. (2019) (https://openreview.net/forum?id=rkluJ2R9KQ). For BCP and ADW, the learned policies are able to match the reference policies; for OKD, the learned policy can significantly outperform the reference policy.

---

> > ### Comment · Reviewer_d4qr · 2023-09-25
> > **response to authors**
> >
> > - For the theoretical result, can you explain why \kappa is tight up to a constant of 2?
> > - For the experiment, I do not understand why you claim NPG is guaranteed to converge in a finite number of episode, specially when there is noise in the algorithm. Can you elaborate on that?
> > - Also for the comparison, does your experiments show that reference policy and the RL algorithm have fairly similar performance?

---

> > > ### Author Response · Authors · 2023-10-03
> > >
> > > - **$\kappa$ tight up to $2$:** The proof is in Appendix B. An important equation is on Page 19 that
> > >   $$\kappa_\clubsuit (\theta) = \max_{s \in \mathcal{S}} \frac{d^{\pi_p} (s) \left(\pi^\star (s) (1 - \pi_\theta (s))^2 + (1 - \pi^\star (s)) \pi_\theta (s)^2 \right)}{d^\clubsuit (s) \left(\frac{1}{2} (1 - \pi_\theta (s))^2 + \frac{1}{2} \pi_\theta (s)^2 \right)},$$
> > >   where $\clubsuit$ could be naive or curriculum learning.
> > >   For the upper bound, we take (1) the feature $\phi(s)$ to be orthogonal for different $s$; (2) the learned behavior $\pi_\theta (s)$ to be the opposite of the optimal behavior $\pi_p (s)$, i.e., $\pi_\theta (s) = 1 - \pi_p (s)$. Thus we can get an upper bound of $\kappa \le 2 d^{\pi_p} (s) / d (s)$.
> > >   For the lower bound, we simply take the first step, i.e., $\theta = 0$ and $\pi_\theta (s) = 1/2$, then we have $\kappa \ge d^{\pi_p} (s) / d (s)$. Thus, we can conclude that $\kappa$ is tight up to $2$.
> > > - **Guaranteed convergence:** Indeed the result is a notion of guaranteed convergence for the expected sub-optimality gap, $\mathbb{E} [V^\star - V^t]$. So noise does affect the convergence. We can justify the flat zero curve for baselines by the following intuition: a naive sampling policy will use $2^{\Theta(n)}$ samples to observe a non-zero reward signal, because it should by chance imitate the optimal behavior to reject the first $n/\mathrm{e}$ candidates. Only with non-zero signal could the model update the weights. Thus, it is reasonable that the curve is flat within $10^7$ samples.
> > > - **Performance of reference policies:** As shown in our figures in Section 7 and Appendix C, our RL policies can perform similarly or even better than the reference policies. In particular, for BCP, RL policies are exactly the same as the reference policy because it is the optimal. For other two problems, RL policies are better than the heuristic reference policies.

---

### Review · Reviewer_dyf8 · 2023-09-10

**Summary Of Contributions:**

This paper studies the problem of reinforcement learning to solve combinatorial optimization problems. The proposed method is to use curriculum learning along with a policy optimization procedure to ensure more scalable learning of the optimal policy. Some theory is provided to justify the use of curriculum learning as reducing the distribution mismatch exponentially. The methods and analyzes are applied on 3 combinatorial problems BCP, OKD, and ADW.

**Audience:**

Yes

**Broader Impact Concerns:**

No concerns.

**Claims And Evidence:**

Yes

**Requested Changes:**

My primary recommendation is to fix the clarity issues mentioned above, which will require some significant changes to the writing, but I think it’s most definitely doable for a revision.

Minor suggestions:
- BCP is not really a combinatorial problem. It’s more of an optimal stopping problem. I would note this somewhere where it is introduced such as 3.1
- All the CO problems should have citations to relevant texts/surveys/original papers on the topic.
- In OKD, why not just work with the original variant that gets to observe the reward to optimize? Doesn’t the policy get to see the value anyway at each timestep? It seems like this was artificially made difficult.
- A lot of important information about how the CO problems are formulated – such as state spaces, actions spaces, rewards – are missing from Section 4.1 and have been relegated to Appendix C without a link. I would advise including this information in the main paper. At the very least there should be a reference to the appendix.
- Section 6.2 second sentence: should this be i/n instead of 1/n?
- Section 7 first paragraph is vague and imprecise (see “more than one experiment” and “In one experiment there are more than one training processes”). I would either leave out sentences like this or be more precise or put them where they are relevant.

**Strengths And Weaknesses:**

Overall I like the paper and appreciate the insights. Like all papers, there are weaknesses. However, there are some issues that really detract from the quality of the paper that I hope the authors can fix.

Strengths:
- The paper studies an interesting and important problem. In particular, the continued use of RL to optimize problems in other fields, such as combinatorial optimization, is interesting.
- The results provide formal insight into how curriculum learning can be helpful for policy optimization (exponentially so).
- The results also suggest that methods of prior work can be simplified.

Weaknesses:
- While the results are certainly informative, the paper is very unclear.
    - Algorithm 2 says to construct an easier E’ form E. This is very vague and nonspecific. What is an ‘easier’ environment? Certainly There are many that one can think of that are not very informative and others that are informative but still hard to solve. While I think it’s okay to be nonspecific to maintain generality, the paper does not do a good job of providing concrete examples of elaborating on characteristics that would satisfy this condition. The only part that sheds light on this is Section C.1 in the appendix where it is finally concretely revealed what the curriculum problems look like. This should be discussed when Algorithm 2 is introduced and then concretely stated in Section 7 of the experiments. Overall, it’s important to define the curricula in the main paper.
    - It’s not clear why the problem is formulated as an LMDP, even though it can be. One of the major insights of the LMDP is that they are hard to learn in general, but can be made easier with either hindsight information or some separability. Neither of those are really used to great effect here. Why not model it as a normal MDP/POMDP where you drop some state information? It’s also not clear what gains can be made from such a formulation in theory either since regret bounds for LMDP typically depend polynomial on the number of MDPs, which is exponential for BCD.
    - The lack of history-dependence is a little concerning for some problems. For instance, the BCP has a state that only contains the current position and rank. While this is enough for the worst-case optimal policy, it seems like it was  formulated with prior knowledge that the optimal policy does already look like this rather than letting the algorithm discover this or find a better solution. For certain distributions, you might be able to do a lot better with history-dependence. Related to this, I believe the LMDP paper does use history-dependent policies (contrary to the claim in the present paper in Section 2).
- The results on CO problems do not appear to be very strong compared to reference baselines. It’s not clear that the methods studied here show promise for finding stronger algorithms for specific problem instances or distributions over instances. However, this is not a big deal in my opinion because the insights are still useful and it’s good proof-of-concept.
- There is not a significant algorithm contribution, but the insights are still useful.

---

> ### Author Response · Authors · 2023-09-20
>
> Thank you for your review!
>
> **Weaknesses:**
>
> - **Discussion on the curriculum design:** The design is problem-dependent. For the problems described in our paper (best choice problem, online knapsack, online matching) as well as any similar problems (online load balancing, online set cover, etc.), we can use $n$, the sequence length of online decision-making, to represent the difficulty. For these problems, we construct $E'$ to be the environment with $n$ smaller than that of $E$. While for a generic problem, it suffices as long as the best policy for $E'$ induces a small $\kappa$ in $E$.
>
> - **Why use LMDP:** MDPs cannot represent the task involving a distribution of instances while forcing a single policy. POMDPs are too general for this task and usually have exponential sample complexity. The regret for LMDP is a worst case upper bound. For the same class of online combinatorial optimization problem, a majority of them are very similar, so the dependency on the number of MDPs is usually significantly smaller than the upper bound.
>
> - **Lack of history-dependence:** In Sec 2 we have already said that the optimal policy for LMDP can be history-dependent. However due to time efficiency, we only study history-independent policies. We agree that for some distributions using a history-dependent policy class may yield better results. Our method can extend to be history-dependent naturally (see Sec 6, page 7).
>
> **Requested changes:**
>
> - **BCP as a CO problem:** We follow the statement in Kong et al. (2019) [https://openreview.net/forum?id=rkluJ2R9KQ]. We added a footnote in Sec 3.1 in the revision.
>
> - **Citations for CO problems:** We added citations of original papers in Sec 2 in the revision.
>
> - **OKD variant:** We use the modified variant to show that the optimal policy for a smaller OKD is significantly better at exploration than a naive random policy. Sparse reward underscores the exploration ability.
>
> - **Detailed formulation of online CO problems:** We have provided the reference link in Sec 7 (page 9). In Sec 4.1 we give high level modeling ideas, while the detailed formulation is only for experiment implementations.
>
> - **$i/n$:** This was a typo. We corrected it in the revision.
>
> - **Experiment schedules:** We clarified this in the revision.

---

### Decision · Action_Editor_6hda · 2023-11-02

**Recommendation:** Accept as is

**Comment:**

This paper proposes to combine curriculum learning and policy optimization for reinforcement learning to solve online combinatorial optimization problems. All reviewers think that the problem studied is interesting, and the theoretical results on the BCP problem are insightful, and they all recommended acceptance. These are nice first steps towards understanding the utility of curriculum learning for real-world application of RL, including other online combinatorial optimization problems.

**Audience:**

Yes

**Claims And Evidence:**

Yes